# Rege-1 promotes *C. elegans* survival by modulating IIS and TOR pathways

Yi-Ting Tsai[1], Chen-Hsi Chang[2], Hsin-Yue Tsai[1,3]*

**1** Institute of Molecular Medicine, College of Medicine, National Taiwan University, Taipei, Taiwan, **2** School of Medicine, College of Medicine, National Taiwan University, Taipei, Taiwan, **3** Center of Precision Medicine, College of Medicine, National Taiwan University, Taipei, Taiwan

* hsinyuetsai@ntu.edu.tw

**Data Availability Statement:** All data needed to evaluate the conclusions in the paper are present in the paper and/or the Supplementary Materials. The raw data are also available from GEO under the

## Abstract

Metabolic pathways are known to sense the environmental stimuli and result in physiological adjustments. The responding processes need to be tightly controlled. Here, we show that upon encountering *P. aeruginosa*, *C. elegans* upregulate the transcription factor *ets-4*, but this upregulation is attenuated by the ribonuclease, *rege-1*. As such, mutants with defective REGE-1 ribonuclease activity undergo *ets-4*-dependent early death upon challenge with *P. aeruginosa*. Furthermore, mRNA-seq analysis revealed associated global changes in two key metabolic pathways, the IIS (insulin/IGF signaling) and TOR (target of rapamycin) kinase signaling pathways. In particular, failure to degrade *ets-4* mRNA in activity-defective *rege-1* mutants resulted in upregulation of class II longevity genes, which are suppressed during longevity, and activation of TORC1 kinase signaling pathway. Genetic inhibition of either pathway way was sufficient to abolish the poor survival phenotype in *rege-1* worms. Further analysis of ETS-4 ChIP data from ENCODE and characterization of one upregulated class II gene, *ins-7*, support that the Class II genes are activated by ETS-4. Interestingly, deleting an upregulated Class II gene, *acox-1.5*, a peroxisome β-oxidation enzyme, largely rescues the fat lost phenotype and survival difference between *rege-1* mutants and wild-types. Thus, *rege-1* appears to be crucial for animal survival due to its tight regulation of physiological responses to environmental stimuli. This function is reminiscent of its mammalian ortholog, *Regnase-1*, which modulates the intestinal mTORC1 signaling pathway.

## Author summary

Eukaryotes rely on tightly regulated insulin-IGF signaling and TOR pathways to maintain proper cellular processes such as metabolism, aging, and pathogen defense, which allow them to sense and respond to changes in nutrient availability and environmental stress. While *Regnase-1* is known for its role in regulating immune response in mammals, the importance of its ortholog, *rege-1*, in *C. elegans* was previously unclear. Our study revealed that *rege-1* participates in regulating the TOR signaling pathway, which subsequently affects *C. elegans* lifespan and pathogen defense. By post-transcriptionally regulating the *ets-4* transcription factor, *rege-1* is able to regulate a set of longevity-suppressed genes, including those involved in the peroxisome fatty acid β-oxidation pathway, likely affecting

GSE218235. https://www.ncbi.nlm.nih.gov/geo/query/acc.cgi?acc=GSE218235.

**Funding:** This study was funded by grants to H.Y. T. from the Ministry of Science and Technology (MOST) of Taiwan (108-2320-B-002-075-MY3, 110-2634-F-002-044), the Ministry of Education in Taiwan, National Taiwan University (110L901402B), and National Taiwan University (110L893404, 111L892804, 112L891604). The funders did not play a role in the study design, data collection and analysis, decision to publish, or preparation of the manuscript. Y.T.T. received her salary from National Taiwan University (110L893404, 111L892804, 112L891604).

**Competing interests:** The authors have declared that no competing interests exist.

the level of acetyl-COA and modulating the TOR signaling pathway. Our findings indicate that defective *rege-1* leads to an excess of *ets-4*, which activates the lipid β-oxidation pathway and further activates the TOR pathway, ultimately reducing lifespan and survival through pathogen and fat loss.

## Introduction

Metabolic pathways have long been known respond to environmental stimuli, such as nutrition sources or environmental stressors. In turn, these metabolic alterations can have major impacts on the organism. For example, suppressing either of two major metabolic pathways in *C. elegans*–insulin/insulin-like growth factor signaling (IIS) or target of rapamycin (TOR) kinase signaling pathways–can extend lifespan, alleviate harmful stress responses, and stimulate other protective mechanisms [1–4]. Mutation of *daf-2*, the only IIS pathway receptor ortholog in *C. elegans*, upregulates lifespan extension and pathogen defense processes mediated by the FOXO transcription factor, *daf-16* [2,3]. Two classes of longevity regulatory genes have been defined, with longevity-associated genes that are upregulated comprising class I and repressed genes belonging to class II (e.g., *ins-7*) [5]. Chromatin immunoprecipitation sequencing (CHIP-seq) analysis has revealed that class I genes are mostly bound by DAF-16 and a minority by PQM-1, the two transcription factors, whereas promoters of class II genes are mostly bound by PQM-1 [6]. While both DAF-16 and PQM-1 promote the activation of class I genes and are required for *daf-2* dependent longevity, class II genes are activated by PQM-1 and their repression is dependent on DAF-16 likely through exclusion of PQM-1 from the nucleus [6].

The TOR kinase pathway is crucial for regulating cell growth by interpreting signals related to nutrient availability [7]. The main complex involved in this process is TOR complex 1 (TORC1), which is activated by both phosphorylation and acetylation upon detection of nutrients. Akt, a protein kinase, is responsible for phosphorylating TOR kinase, while EP300, an acetyltransferase, acetylates Raptor, a key adaptor protein in the TORC1 [8]. The activation of TORC1 kinase activity also requires the GTP-bound RHEB, and the heterodimeric Rag GTPase (composed of RAGA and RAGC) is essential for recruiting activated TORC1 to lysosomes to continue the TORC1 signaling cascade [7]. This cascade enhances the phosphorylation of S6 ribosomal proteins and 4EBP1, further promoting translation. Additionally, the activation of TORC1 signaling inhibits the activation of autophagy by preventing the formation of the ULK complex, the initiation complex [9]

The level of acetyl-CoA plays a crucial role in the acetylation of Raptor [10]. In amino acid-starved cells, TORC1 signaling is activated by supplementing with acetyl-CoA. Acetyl-CoA can be derived from various nutrients such as pyruvate, amino acids, and fatty acid β-oxidation [8]. However, a study on modulating fatty liver formation in fasting or high fat diet mice has shown that blocking the liver peroxisome fatty acid β-oxidation pathway by deleting acyl-CoA Oxidase 1 (ACOX1) is sufficient to reduce fatty liver due to decreases in acetyl-CoA, which consequently results in decreased TORC1 signaling and activates autophagy [8,10].

Although most research on the regulation of metabolic pathways has focused on transcriptional control and post-translational modifications, post-transcriptional regulations, which is primarily mediated by microRNAs and their associated Argonautes, have been found to regulate metabolic pathways [11]. For instance, *miR-100* can prevent the activation of TORC1 [12], and *let-7a* targets the insulin receptor, reducing insulin signaling [13]. However, unlike Argonautes that identify specific mRNA targets through microRNAs, most endo- and exo-

ribonucleases regulate mRNA degradation through general mRNA decay pathways [14]. Nonetheless, Regnase-1 is a unique endoribonuclease that targets selected mRNAs, mostly in their 3' UTR, in different cell types, including those that encode several cytokines in macrophages, the transcription factor BATF in T cells, and mTOR signaling pathway proteins in the intestine [15–17].

Regnase-1 contains a ZC3h12A-like ribonuclease domain and a CCCH zinc finger domain and is only present in multicellular eukaryotes, with no homologs found in bacteria or fungi [18]. In *C. elegans*, the only Regnase-1 protein ortholog is REGE-1, which specifically targets *ets-4* mRNA, an *ets* (E-twenty-six-specific sequence) transcription factor [19,20]. It is worth noting that while *ets* genes are widely distributed among metazoans, no *ets* genes have been identified in protozoans [21]. Previous research has demonstrated that *rege-1* mutants exhibit a significant loss of fat, and mRNAseq analysis has revealed the upregulation of lipid catabolic and immune-related genes [22]. Interestingly, genetic studies have shown that the combination of an *ets-4* mutation and a *rege-1* mutation is sufficient to restore normal fat levels and prevent immune gene misregulation. However, how these two newly evolved metazoan proteins participate in *C. elegans* cellular processes and survival remains largely unknown. Specifically, whether an excess of ETS-4 results in detrimental effects in *C. elegans* and needs to be regulated by REGE-1 is still yet to be determined.

ETS-4 contains both an ETS DNA binding domain and a PNT domain. Previous studies have shown that deleting *ets-4* leads to a longer lifespan and increased cold tolerance [23,24]. However, ETS-4 is not always associated with negative effects in *C. elegans*, as it has been found to be essential for axon regeneration by promoting the expression of a receptor tyrosine kinase, SVH-2 [25]. While ETS proteins are capable of acting as either transcriptional activators or repressors, *ets-4* has been shown to primarily function as a transcriptional activator, as evidenced by studies in yeast and mouse fibroblast reporter systems [24]. Additionally, the localization of ETS-4 appears to be dynamic. While ETS-4 is observed in the nucleus of neuron, pharyngeal and embryonic intestinal cells, it is less frequent in the nucleus of adult intestinal cells [24]. However, the nuclear localization of ETS-4 is increased in adult intestinal cells during cold stress or in the presence of a *rege-1* mutant background [22,23].

Similar to *rege-1* mutant in *C. elegans*, total and mobile immune cell (T cell, B cell and macrophage)-specific *Regnase-1* knockout mice result in strong up-regulation in proinflammatory cytokines and develop autoimmune syndromes [16,26–28]. The 50% death rate for total *Regenase-1* knockout is within eight weeks. Intriguingly, mice with lung- or intestine-specific *Regnase-1* knockouts show even better survival than wild-type mice when challenged with *P. aeruginosa* and dextran sulfate sodium (DSS), respectively [17,29]. Studies have demonstrated that the *Regnase-1* deficits increase *P. aeruginosa*-specific IgA levels in bronchoalveolar lavage fluid or enhance intestinal cell proliferation by activating the mTOR signaling pathway, providing explanations for their better survival. [17,29].

In this study, we demonstrated that *rege-1* mutant worms have a shorter lifespan than wild-types, which is further exacerbated upon exposure to *P. aeruginosa* (PA14). Our mRNA-seq results suggest that the decreased survival after PA14 exposure is due to changes in both the IIS and TOR kinase signaling pathways, rather than the known immune pathways in *C. elegans*. Survival analysis after PA14 exposure revealed that genetic inhibition of either the initiation of the IIS or TORC1 pathway fully rescues the survival defect in *rege-1* mutants, and the fat content can be rescued in genes related to the IIS pathway. The mRNA-seq and ETS-4 ChIP-seq analysis revealed that class II genes in the IIS pathway, but not TORC1, exhibit the upregulated gene expression in *rege-1* mutant and are also targeted by ETS-4. We further examined potential genes that link the class II genes and the TORC1 pathway and discovered that the *acox-1.5* mutant, one of the peroxisome acyl-coA oxidase 1 proteins, is the best rescue

candidate for the survival and fat difference between wild-type and *rege-1* mutant. Overall, our results demonstrate that *rege-1* affects *C. elegans* survival by fine-tuning two key metabolic control systems.

## Results

### REGE-1 ribonuclease activity is required for *C. elegans* lifespan and survival upon *P. aeruginosa* exposure

To investigate the role of REGE-1 in *C. elegans* survival, we assessed the lifespan of *rege-1 (tm2265)*, a deletion strain lacking the entire 4th exon resulting in a frameshift in the coding sequence. As the Zc3h12a NYN-like ribonuclease domain of REGE-1 spans exons 4 to 7, the *rege-1(tm2265)* strain only retains the first 147 amino acids of the wild-type REGE-1 protein and lacks the ribonuclease domain (Fig 1A). To test the effect of REGE-1 on pathogen response, we exposed the *rege-1(tm2265)* to PA14, which can infect both humans and *C. elegans*. Compared to the wild-type strain, we observed significant decreases in both lifespan at 20°C (83.5% relative to N2 mean lifespan) and survival after PA14 exposure (66.8% relative to N2 mean lifespan) (Fig 1B, 1C and S1 Table). As REGE-1 is a ribonuclease, we generated a ribonuclease-defective mutant, *rege-1(imm070)*, by altering the aspartic acid residue at position 231 to asparagine, which has been shown to abolish the ribonuclease activity of ZC3h12A-like ribonuclease domain proteins in *C. elegans* and mouse (Fig 1A, [16,30,31]). Both the deletion mutants and the ribonuclease-defective mutants had notably shorter mean lifespans than N2, with reductions of 82.0% and 83.5% in *rege-1(imm070)* and *rege-1(tm2265)*, respectively. Moreover, their survival rates after exposure to PA14 were poor, with declines of 71.3% and 66.8% in *rege-1(imm070)* and *rege-1(tm2265)* relative to N2 mean lifespan. These findings suggest that the ribonuclease activity of REGE-1 plays a crucial role in the survival of *C. elegans*. (Fig 1D, 1E and S1 Table).

Since ETS-4 mRNA is known to be a key substrate of REGE-1, we measured both lifespan and survival after PA14 exposure in both *rege-1* mutants with *ets-4(ok165)* (Fig 1A–1E and S1 Table). The *ets-4(ok165)* strain, which the entire PNT and ETS domains have been deleted, was reported to exhibit an extended lifespan [24], and we did observe a lifespan extension of *ets-4(ok165)* when fed with OP50 (120.4% relative to N2 mean lifespan). However, we found no significant difference in the survival curves of *ets-4(ok165)* and wild-type worms after exposure to PA14 (Fig 1B–1E and S1 Table). Importantly, comparable PA14 survival curves were observed among wild-type and the two *rege-1; ets-4* strains (Fig 1C, 1E and S1 Table). Since a previous study of *ets-4* suggests that it acts as a transcriptional activator, we hypothesized that the main cause of PA14-shortened lifespan in *rege-1(imm070)* could be the upregulation of a set of tightly regulated genes by excess ETS-4. To begin testing this hypothesis, we quantified the mRNA levels of both *ets-4* and *rege-1* after feeding with OP50 or PA14 in all four strains: N2, *rege-1(imm070)*, *ets-4(ok165)*, and *rege-1(imm070); ets-4(ok165)*. Interestingly, not only *ets-4* was up-regulated upon PA14 exposure, we also found a high level of *ets-4* mRNA in *rege-1(imm070)* mutants in either OP50 or PA14 fed samples, suggesting not only our single amino acid substitution in *rege-1(imm070)* is sufficient to abolishing the ribonuclease activity as expected, an excess of *ets-4* is a key factor for *rege-1*-dependent sensitivity of *C. elegans* to PA14 (Fig 1F).

To gain further insight into the factors contributing to the poor survival of *rege-1* on PA14, we investigated potential differences in pathogen clearance rates and avoidance among all four strains, N2, *rege-1(imm070)*, *ets-4(ok165)*, and *rege-1(imm070); ets-4(ok165)*. Our findings indicate that no observable difference exists in the rate of RFP-labeled PA14 clearance between the wild type (N2) and *rege-1 (imm070)* after transferring the worms from 24 hours of

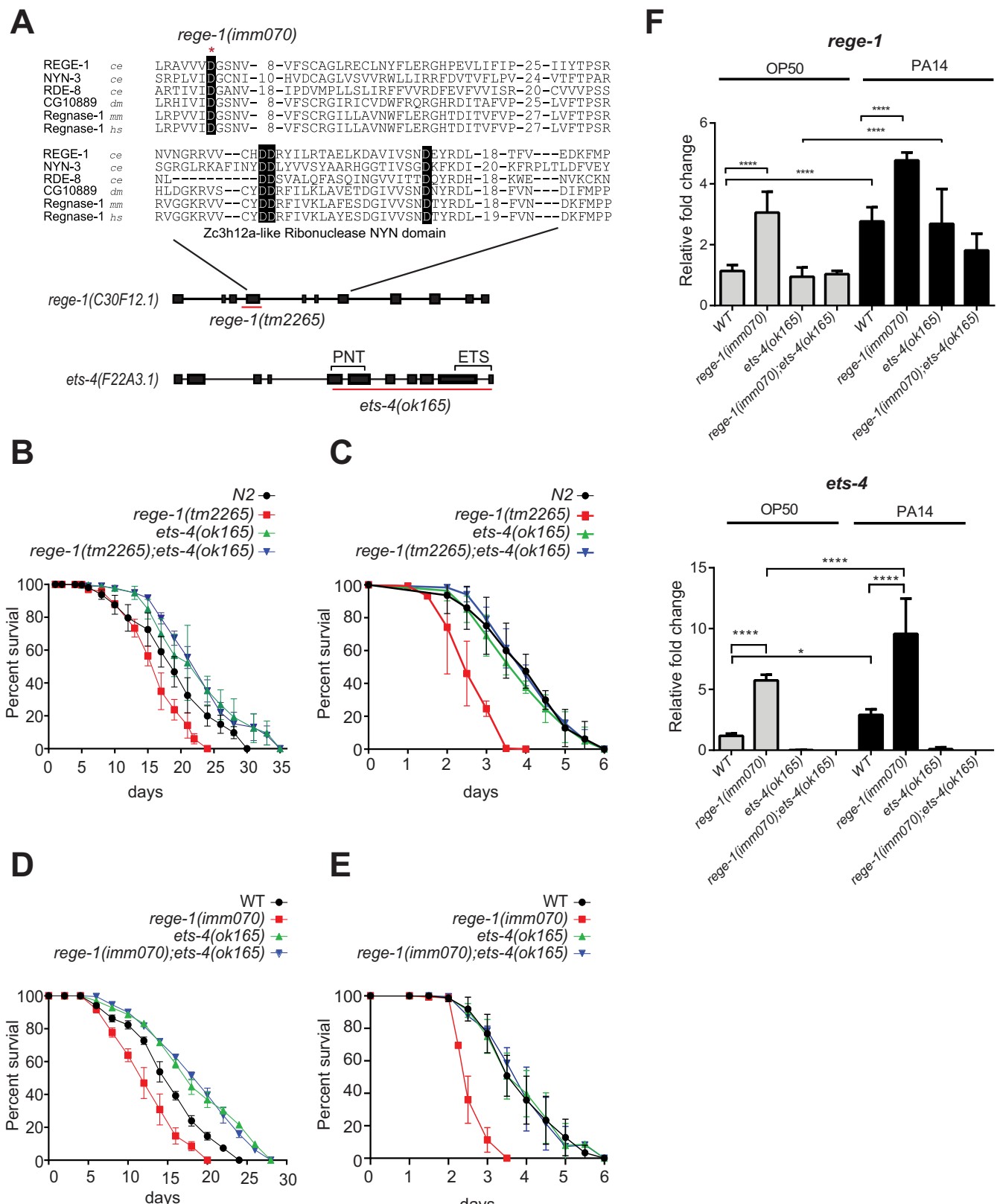

**Fig 1. REGE-1 is important for *C. elegans* lifespan and survival under PA14 through modulating *ets-4*.** (A) Schematic diagram showing the gene structure of *rege-1* and *ets-4*, with black boxes representing exons and black lines representing introns. The alignment of Zc3h12a-like ribonuclease NYN domain from

different species is presented, and the PNT and ETS domains of ETS-4 are indicated. Species abbreviations: ce, *Caenorhabditis elegans*; dm, *Drosophila melanogaster*; mm, *Mus musculus*; hs, *Homo sapiens*. Conserved aspartic acid residues are highlighted with a black background. The red asterisks denote the point mutations in *rege-1 (imm070)*, while the red lines show the deletion in *rege-1 (tm2265)* and *ets-4(ok165)*. (B, D) Lifespan analysis of N2, *rege-1* mutants, *ets-4(ok165)*, and *rege-1;ets-4(ok165)* double mutants at 20°C on OP50. The *rege-1* mutant is denoted as *rege-1(tm2265)* in (B) and *rege-1(imm070)* in (D). (C, E) PA14 survival curve for N2, *rege-1* mutants, *ets-4(ok165)*, and *rege-1;ets-4(ok165)* double mutants on PA14 at 25°C are presented. The *rege-1* mutant is denoted as *rege-1(tm2265)* in (C) and *rege-1(imm070)* in (E). The Error bars represent SEM was shown in survival curve. Three repeats were done in all survival curves and the raw data are present in S1 Table. (F) The mRNA levels of *rege-1* and *ets-4* were measured by RT-qPCR with *act-3* mRNA as an internal control. The fold changes relative to wild-type animals fed by OP50 were shown. (n = 3, * p< 0.05, **** P < 0.0001, Error bar represent SEM, One-way ANOVA with Tukey's multiple comparison test).

PA14-fed to regular OP50 food for one day (S1A and S1B Fig). Additionally, we assessed the PA14 avoidance behavior of all four strains, finding that over 80% of the worms evaded PA14 after 24 hours (S1C Fig.). There was no significant difference between the four strains tested, suggesting that the PA14 sensitivity of *rege-1(imm070)* is unlikely due to the failure of clearing or escaping from the pathogen.

### *Rege-1* negatively regulates genes involved in IIS and TOR signaling pathways

To investigate the potential mechanism by which excess *ets-4* could lead to reduced survival following PA14 exposure, we conducted mRNA-seq analysis on three strains of day 2 adult worms that were fed either OP50 or PA14: N2, *rege-1(imm070)*, and *rege-1(imm070);ets-4 (ok165)*. Since the phenotype analysis showed that only *rege-1(imm070)* had a significantly decreased survival rate, while *rege-1(imm070);ets-4(ok165)* had a similar lifespan to wild-type when fed with PA14, we reasoned that only the genes that consistently differed between *rege-1 (imm070)* and the other two strains are likely to contribute to the poor survival phenotype. Thus, we narrowed our search to 173 and 138 differentially expressed genes (DEGs) in OP50- and PA14-fed samples, respectively (S2A and S2B Fig and S2 Table). Consistent with the previous finding that ETS-4 is a transcription activator, we found that the majority of DEGs (78% and 64% in OP50- and PA14-fed samples, respectively) were upregulated in *rege-1(imm070)* (Figs 2A and S2C). We then performed enrichment analysis on these two sets of up-regulated genes. Consistent with the intestinal expression of REGE-1, most genes are enriched in the intestine ([22], S2D Fig). The gene ontology analysis shows that, while lipid catabolic processes were found in the up-regulated genes in *rege-1(imm070)* (the REGE-1 negatively regulated genes), immune-related processes are enriched in both up- and down-regulated genes in *rege-1(imm070)* (S2D Fig).

To determine if the set of genes negatively regulated by *rege-1* is associated with other transcriptomic datasets resulting from mutations in other genes, we utilized WormExp v2.0 specifically on mutant datasets [32]. As expected, the categories "UP by *rege-1* RNAi" and "down by *ets-4* RNAi" were the most significant. Surprisingly, the pathways involved in energy metabolism, namely "down by *daf-2;rsks-1* mutant" and "down by *aak-2* overexpressed and *daf-2* mutant," were also found to be significant (Table 1). These two datasets were obtained by examining the synergistic effects on lifespan by reducing IIS and TOR pathways, and AAK-2/ IIS/TOR signaling pathways participation in dietary restricted longevity, respectively. To narrow our search, we correlated our datasets with the "DAF/Insulin/food" category in WormExp and found that the top five lists that overlapped with *rege-1* negatively regulated genes were either individual or a combination of IIS and TOR signaling pathways (Table 1, [33,34]).

To understand whether the *rege-1* negatively regulated genes correlate with differential gene expression (DGE) in *daf-2* mutants, we compared the gene lists with previously identified

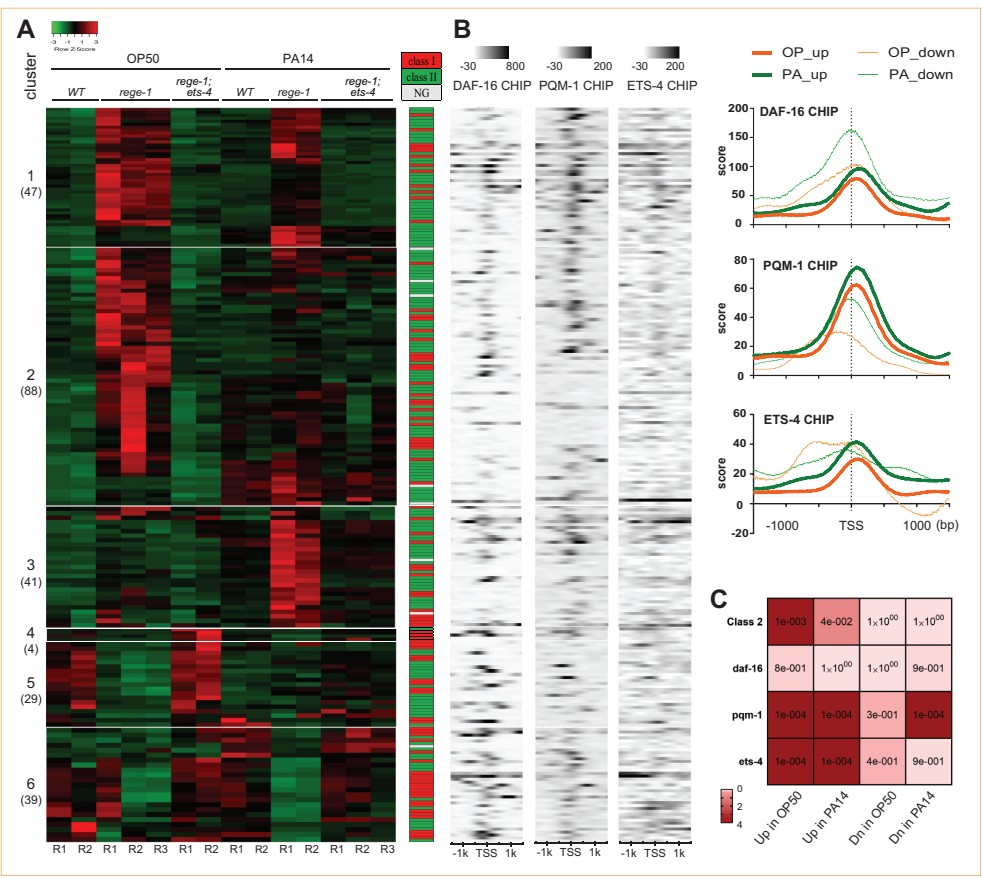

**Fig 2. *Rege-1* negatively regulated gene are enriched in class II genes and bound by ETS-4 and PQM-1.** (A) Heat maps and clusters of the DEGs from OP50- and PA14-fed samples are shown. The DEGs were classified into six clusters based on their expression dependency of *rege-1(imm070)*: Cluster 1, genes up-regulated in *rege-1(imm070)* in both OP50- and PA14-fed samples; Cluster 2, genes up-regulated in *rege-1(imm070)* in OP50-fed samples only; Cluster 3, genes up-regulated in *rege-1(imm070)* in PA14-fed samples only; Cluster 4, genes down-regulated in *rege-1(imm070)* in both OP50- and PA14-fed samples; Cluster 5, genes down-regulated in *rege-1(imm070)* in OP50-fed samples only; and Cluster 6, genes down-regulated in *rege-1(imm070)* in PA14-fed samples only. The number of DEGs in each cluster is indicated in parentheses. Genes belonging to either class I or class II were color-coded on the right, with red for class I, green for class II, and gray for genes without a specified class in the dataset from Tepper et al., Cell (2013). (B) A heatmap displays the density of ChIP-seq data of DAF-16, PQM-1, and ETS-4 within 1.5k bp of transcription start site (TSS) corresponding to gene listed in the expression heatmap on the left. All ChIP data were extracted from the ENCODE project. Right: Accumulative plots of ChIP-seq data of positively and negatively regulated genes by *rege-1(imm070)* in either OP50- or PA14-fed samples. (C) The statistical analysis of DEGs of the *rege-1(imm070)* overlapping with PQM-1, DAF-16, and ETS-4 targeted genes is shown."Up" and "Dn" denote as up- and down-regulated genes in *rege-1(imm070)*, respectively. "OP" and "PA" denote as worms were fed with OP50 and PA14, respectively.

class I and class II genes, DAF-16, and PQM-1 ChIP data and also the ETS-4 ChIP data from the ENCODE database (Fig 2B, [6,35]). A total of 69.6% (94/135) and 68.2% (60/88) of *rege-1* negatively regulated genes from OP50- and PA14-fed, respectively, were found to be significantly enriched in class II, whereas no significant enrichment in either class was found in *rege-1* positively regulated genes in either group (Fig 2C). We then further examined the ETS-4, PQM-1, and DAF-16 occupancies on the *rege-1* DEGs (Fig 2B). Although ETS-4 occupancy is significantly enriched in *rege-1* negatively regulated genes, only 15.6% (21/135) and 20.5% (18/88) genes in OP50- and PA14-fed gene sets, respectively, were found bound by ETS-4 (S3 Table). We reason that the experimental condition difference between the ENCODE

**Table 1. Overlapping *rege-1* negatively regulated genes in OP50 and PA14 with existing datasets.**

| *Rege-1* negatively regulated genes fed with OP50 | |
| --- | --- |
| **Mutant datasets** | **FDR** |
| UP by *rege-1* RNAi (Habacher) | 1.21E-92 |
| down by *ets-4* RNAi (Habacher) | 2.97E-86 |
| UP by *tax-4* RNAi | 1.81E-31 |
| down by *daf-2;rsks-1* mutant | 9.52E-30 |
| down by *aak-2* overexpressed and *daf-2* mutant | 7.80E-24 |
| **DAF/Insulin/food datasets** | **FDR** |
| UP by *daf-16* mutant under *daf-2;rsks-1* mutant | 3.23E-32 |
| down by *daf-2* mutant (Chen) | 1.52E-27 |
| Genes upregulated by DR | 1.47E-26 |
| Down N2-Cel-RhebRNAi-fasting | 7.55E-25 |
| down by *daf-2* mutant (Knutson) | 4.24E-20 |
| *Rege-1* negatively regulated genes fed with PA14 | |
| **Mutant datasets** | **FDR** |
| down by *ets-4* RNAi (Habacher) | 9.96E-72 |
| UP by *rege-1* RNAi (Habacher) | 2.17E-70 |
| down by *aak-2* overexpressed and *daf-2* mutant | 3.03E-35 |
| down by *daf-2;rsks-1* mutant | 8.10E-31 |
| Up in *hyl-2* mutants | 2.52E-29 |
| **DAF/Insulin/food datasets** | **FDR** |
| UP by *daf-16* mutant under *daf-2;rsks-1* mutant | 4.50E-29 |
| down by *daf-2* mutant (Chen) | 1.40E-27 |
| Down N2-Cel-RhebRNAi-fasting | 1.94E-27 |
| Down *daf-16* vs. *daf-2* | 1.26E-25 |
| down by *daf-2* mutant (Knutson) | 1.77E-23 |

database and our experiment is likely the reason for the small portions of ETS-4 targets found in *rege-1* negatively regulated gene lists. Moreover, consistent with PQM-1 being the major transcription factor bound to class II genes, the *rege-1* negatively regulated genes in both OP50- and PA14-fed conditions are significantly bound by PQM-1, but not DAF-16 (Fig 2B and 2C). Of note, although there is no significant bias toward either class in *rege-1* positively regulated genes, PQM-1 occupancies are still significantly enriched in PA14-fed *rege-1* positively regulated genes (Fig 2B and 2C). Our analysis results suggest that the *rege-1* negatively regulated genes are significantly enriched in class II genes, and both ETS-4 and PQM-1 are significantly associated with this set of genes.

To understand the similarity of ETS-4 regulated DEGs, we compare DGE between up-regulated ETS-4 due to defective *rege-1*, *rege-1(imm070)* (our data) and *rege-1(rrr13)* as well as *ets-4* mutants, we correlated our dataset with transcriptome analyses with previous datasets [22–24]. As expected, we observed a significant correlation between both ETS-4 positively regulated genes from previous datasets and *rege-1* negatively regulated genes fed by OP50 (S4 Table). Additionally, we found an enrichment of ETS-4 occupancy on *ets-4* positively regulated genes in both previous datasets (S3A Fig).

We next searched for the relevance of *rege-1* negatively regulated genes and the TOR signaling pathway [36]. Based on the results from WormExp, *rege-1* negatively regulated genes are enriched in down-regulated genes in *daf-2;rsks-1* mutant. To determine whether the significance is only restricted to *rege-1* negatively regulated genes like IIS pathway or if it is a global

**Table 2. The pearson correlation of the *rege-1(imm070)* differential expressed genes with daf-2(e1370);rsks-1 (ok1255) *in OP50 and PA14.***

| OP50 | | |
|---|---|---|
| *daf-2(e1370); rsks-1(ok1255)* | *rege-1(imm070)* vs WT | *rege-1(imm070)* vs *rege-1(imm070);ets-4(ok165)* |
| Pearson r | -0.1847 | -0.2628 |
| P (two-tailed) | 0.0278 | <0.0001 |
| P value summary | * | **** |
| Number of XY Pairs | 142 | 447 |
| **PA14** | | |
| *daf-2(e1370); rsks-1(ok1255)* | *rege-1(imm070)* vs WT | *rege-1(imm070)* vs *rege-1(imm070);ets-4(ok165)* |
| Pearson r | -0.4367 | -0.2548 |
| P (two-tailed) | <0.0001 | <0.0001 |
| P value summary | **** | **** |
| Number of XY Pairs | 135 | 284 |

inverse correlation, we analyzed the global DGEs between *daf-2(e1370);rsks-1(ok1255)* and *rege-1(imm070)* in either OP50- or PA14-fed conditions (Tables 2 and S5). Interestingly, we found a significant inverse correlation of all our data sets to *daf-2(e1370);rsks-1(ok1255)* DGE. RSKS-1 is a *C. elegans* ortholog of p70-S6 kinase 1, which is known downstream of the TOR signaling pathway to promote translation by phosphorylating S6 ribosomal protein. To determine whether the global mRNA changes mimicking the inverse correlation of decreases in TOR signaling in *rege-1(imm070)* is due to a specific TOR pathway gene being up-regulated, we identified gene expression profiles from mRNA-seq of all the *C. elegans* orthologs found in the TOR pathway, core components, interactors, substrates, or regulators. Surprisingly, no statistically significant difference was found between *rege-1(imm070)* and the relative control groups (S2C Fig). We further checked whether there is an enrichment in ETS-4 occupancy in down-regulated genes in *rsks-1;daf-2* mutant relative to wild-type that overlapped with *rege-1* negatively regulated genes. No significant ETS-4 occupancy was found (S3B Fig). Our data analysis suggests that the poor survival observed in *rege-1(imm070)*, both in OP50- and PA14-fed conditions, is likely due to the activation of the TOR signaling and class II genes. The activation of class II genes is likely due to direct binding of ETS-4, whereas activating TOR signaling is less likely due to ETS-4 activating gene expression of core TOR signaling pathway.

## The *daf-2* mutant, but not the *pqm-1* mutant, is sufficient to rescue the PA14 sensitivity of *rege-1(imm070)*

To validate our sequencing data analysis, we investigated whether decreasing class II gene expression could rescue the poor PA14 survival phenotype of *rege-1(imm070)*. We introduced two strains: *daf-2(e1370)*, which contains a single amino acid change in the kinase domain resulting in strong dauer formation and extended lifespan, and *pqm-1(tm8184)*, which carries a 96 bp deletion at the splicing donor site in exon 3 resulting in the loss of the C2H2 zinc-finger DNA binding domain. *Daf-2(e1370)* is known to repress class II gene expression and enhance class I gene expression, while PQM-1 is a transcription factor that specifically targets class II genes [5,6]. Since 56% of ETS-4 targeted genes from ETS-4 ChIP data were also classified as PQM-1 targeted genes, we further investigated whether PQM-1 and ETS-4 are

functionally dependent on each other. Specifically, we tested whether the absence of PQM-1 alone could suppress the phenotype caused by an excess of ETS-4.

Our PA14 survival assay demonstrated that introducing *daf-2(e1370)* into the *rege-1 (imm070)* background increased the mean lifespan by 1.5-fold compared to N2. No significant difference was observed between *daf-2(e1370)* and *rege-1(imm070); daf-2(e1370)* (Fig 3A and S1 Table). However, the deletion of *pqm-1* had no effect on poor PA14 survival or shorter lifespan in OP50-fed conditions in *rege-1(imm070)* (Fig 3B and 3C). We defined mean lifespan as having no significant difference from *rege-1(imm070)* as 0% rescue, and mean lifespan having no significant difference from the introduced mutant as 100% rescue. Based on this calculation, our results suggest that abolishing IIS using *daf-2(e1370)* resulted in a 100% rescue of the poor survival phenotype on PA14 in *rege-1(imm070)*. However, even though ETS-4 and PQM-1 both target class II genes, missing PQM-1 did not affect the poor PA14 survival or short lifespan phenotype in *rege-1(imm070)* (0% rescue).

### ETS-4::GFP localization depends on the presence of DAF-16

It is well known that PQM-1 nuclear localization is inhibited by DAF-16 in *daf-2* mutant. We also tested whether ETS-4 localization depends on DAF-16. A *ets-4::gfp* construct containing *ets-4* promoter and coding sequence with *unc-54* 3' UTR was co-injecting with the dominant negative *rol-6* plasmids. The ratio of GFP nuclear intensity was quantified in both wild-type and *daf-16(mu86)* strains. Although only a slightly higher nuclear localized ETS-4::GFP was found in wild-type, when compare to autofluorescence in N2 strain without ETS-4::GFP, a much stronger nuclear ETS-4::GFP signal was found in *daf-16(mu86)* compared to wild-type. Our results suggest that, although PQM-1 nuclear exclusion by DAF-16 occurs only in *daf-2* mutants but not under wild-type conditions, ETS-4 nuclear localization is hindered by DAF-16 even under wild-type conditions. (Fig 3D, [6,23]).

### Activation of the class II gene, *ins-7*, promoter in *rege-1(imm070)*

To further confirm our analysis that the promoter of a class II gene is targeted by ETS-4, we selected *ins-7*, a gene that is targeted by ETS-4 in ETS-4 ChIP-seq and negatively regulated by *rege-1* (Fig 3E and 3F). *Ins-7* encodes one of the forty insulin peptides predicted to act as ligands for the DAF-2 insulin receptor in *C. elegans*. It is well known that *ins-7* expression is repressed with reduced DAF-2 activity [5]. We first confirmed that *ins-7* expression is increased in *rege-1(imm070)* as observed from our mRNA-seq results (Fig 3F). To investigate whether *ins-7* transcription is indeed promoted by ETS-4, we incorporated a GFP reporter driven by the *ins-7* promoter (*pins-7::gfp*) in *rege-1(imm070)* [37]. By quantifying the GFP expression in day 2 adults, we found that GFP expression in *rege-1(imm070)* was enhanced relative to wild-type worms (Fig 3G). These results suggest that *ins-7*, one of the class II genes, is transcriptionally upregulated in *rege-1(imm070)* due to excess ETS-4.

### Suppressing TORC1 signaling rescues survival of PA14-fed *rege-1(imm070)*

We next investigate the possible involvement of the TOR signaling pathway in the decreased survival of PA14-exposed *rege-1(imm070)*, and therefore, we assessed the survival rates of worms that had been genetically manipulated to inhibit two *C. elegans* TOR signaling pathway orthologs, *raga-1* and *rsks-1*. As noted earlier, *raga-1* is the ortholog of RAGA, and *rsks-1* is the ortholog of p70-S6 kinase 1. They represent the steps involved in recruiting activated TORC1 to lysosomes and mediating translation activation through active TORC1 signaling, respectively. *Raga-1(ok386)* has a large deletion that removes exon 2–4 of *raga-1*, whereas *rsks-1 (tm1714)* carries a 484bp deletion in exon 4, resulting in a frameshift in the coding sequence.

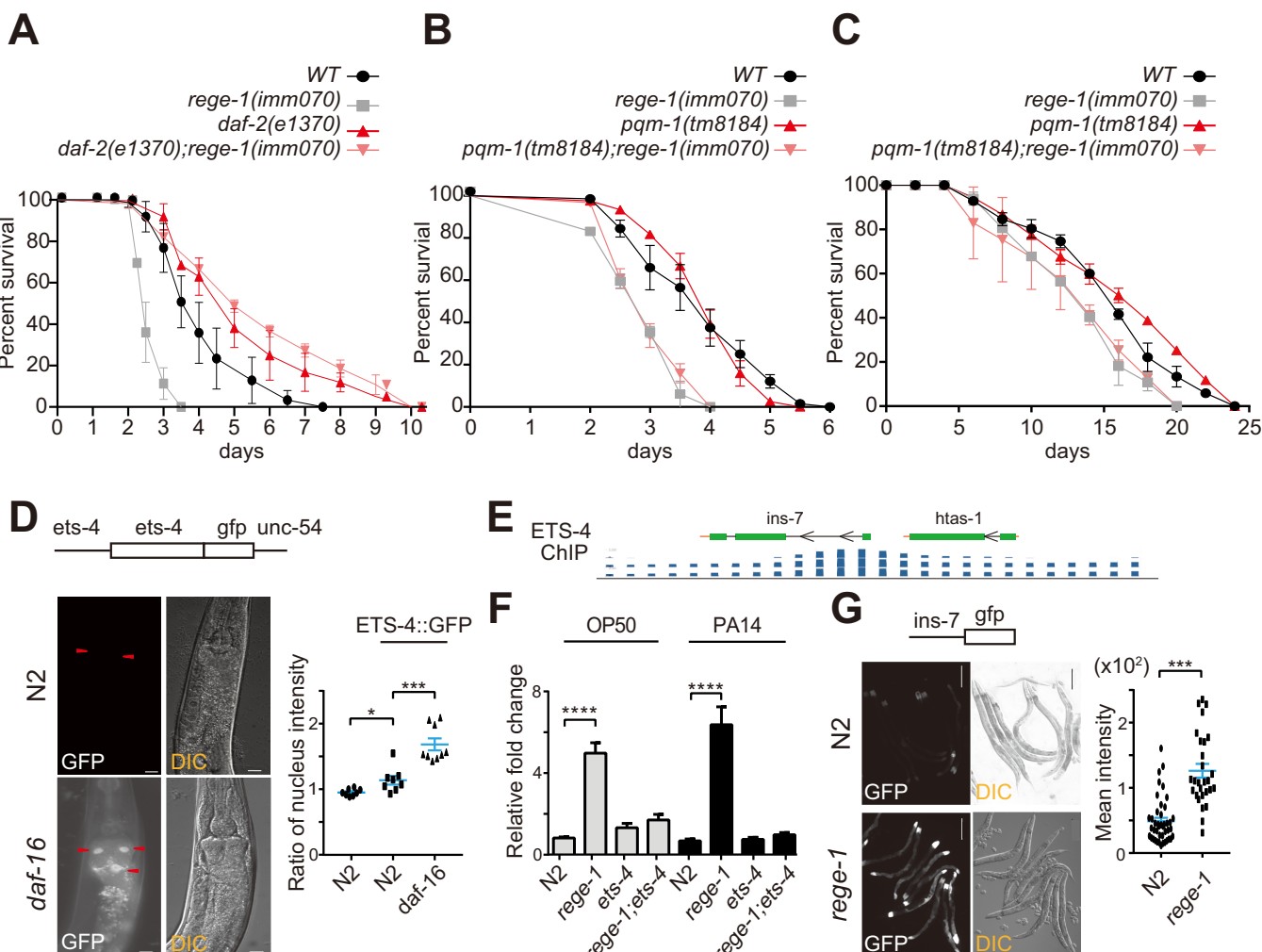

**Fig 3. The IIS pathway genes contributes to the PA14 sensitivity of *rege1(imm070)*.** (A,B) PA14 survival curves of wild-type (WT), *rege-1(imm070)* and (A) *daf-2(e1370)* and *rege-1(imm070);daf-2(e1370)*, (B) *pqm-1(tm8184)* and *rege-1(imm070)*; *pqm-1(tm8184)*. (C) Lifespan of wild-type (WT), *rege-1(imm070)*, *pqm-1(tm8184)* and *rege-1(imm070)*; *pqm-1(tm8184)* at 20˚C. Three and two replicates have been performed in all the PA14 survival curves and lifespan, respectively. (D) Ratio of ETS-4::GFP nuclear localization in N2 and *daf-16(mu86)*. The extrachromosomal array of *ets-4::gfp* was injected into either N2 or *daf-16(mu86)* strains, and the GFP intensity was quantified as described in the methods. Due to strong autofluorescence background, N2 worms without *ets-4::gfp* injection was also quantified as N2 without ETS-4::GFP. Worms with successful plasmid injection was scored by containing dominate negative *rol-6* plasmid which shows a strong roller phenotype. The relative nuclear GFP intensity ratio is shown on the right. This ratio was calculated as the intensity of the nucleus divided by the intensity of the intestinal cell and was further normalized with the N2 strain without the *ets-4::gfp* plasmid injection. The white scale bars in the corner of the images represent a length of 10 μm. Two to three nuclei were quantified in each worm, and statistical significance between samples was determined using a one-way ANOVA with Turkey's multiple comparison test. (E) Genome browser displaying ETS-4 ChIP-seq data on *ins-7*. The exon of the gene is denoted as a green box, the intron is shown as a black line, the orange line indicates the 3' or 5' UTR defined in the genome browser, and the black arrows indicate the direction of the gene. The blue lines below the gene structures represent the read intensity identified from ETS-4 ChIP-seq data. (F) Quantification of *ins-7* mRNA in wild-type, *rege-1(imm070)*, *ets-4(ok165)*, and *rege-1(imm070);ets-4(ok165)* fed with OP50 or PA14. Three biological replicates were performed for each sample. Statistical analysis was performed using one-way ANOVA with Tukey's multiple comparison test. (G) GFP expression was analyzed in *pins-7::gfp* strains in either N2 or *rege-1(imm070)* genetic backgrounds. Representative images are shown on the left, and quantification of GFP intensity is shown on the right. Significant differences between samples were calculated using an unpaired t-test. The scale bar in the corner of the image represents a length of 200 μm. Each experiment was scored with 15–20 worms, and two independent repeats were performed for each experiment. (*p < 0.05, **p < 0.001, ***p < 0.0002, ****p < 0.0001).

Excitingly, we found that the decrease in PA14 survival in *rege-1(imm070)* was completely rescued to wild-type mean lifespan when either *raga-1(ok386)* or *rsks-1(tm1714)* was introduced to the *rege-1(imm070)* genetic background (Fig 4A, 4B and S1 Table). No significant difference

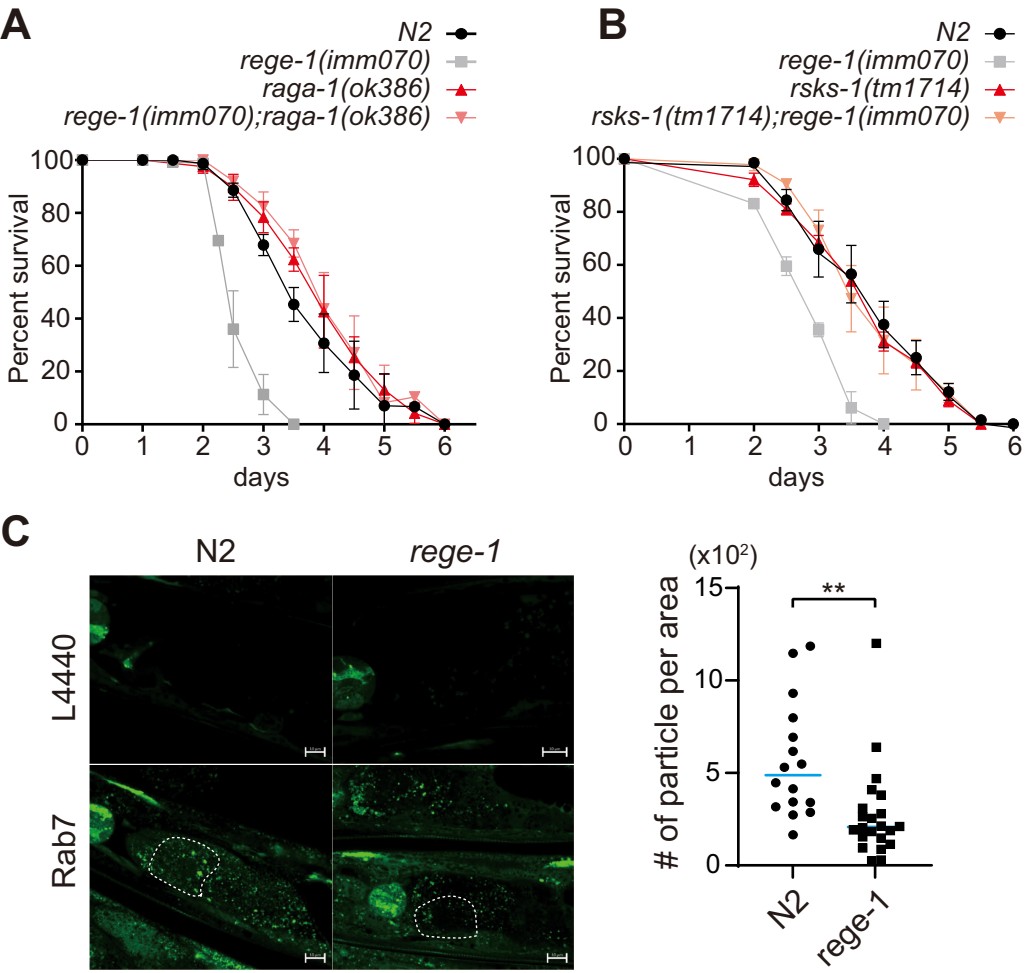

**Fig 4. Suppressing TORC1 signaling rescues survival of PA14-fed *rege-1(imm070)*.** (A,B) PA14 survival curves of wild-type, *rege-1(imm070)* and (A) *raga-1(ok386)* and *rege-1(imm070); raga-1(ok386)*. (B) *rsks-1(tm1714)* and *rege-1 (imm070); rsks-1(tm1714)*. Three replicates have been performed in all survival curves, and raw data and significance is shown in S1 Table (C) Autophagosome formation was quantified in *N2* and *rege-1(imm070)* strains. Worms containing the *gfp::lgg-1* transgene were crossed into both strains. The worms were fed with either L4440 as an RNAi control or *rab-7* RNAi from the L1 larval stage, and GFP imaging was performed on day 2 of adulthood. The detailed quantification procedures are described in the Materials and Methods section. An unpaired t-test was used to quantify the difference in the amount of autophagosome formation. The white dot lines indicate an example of a cell circle for quantification.

of mean lifespan was observed between N2, *raga-1(ok386)* and *rsks-1(tm1714)*. Our results are consistent with mRNA-seq analysis indicating that TORC1 signaling activation likely occurs in *rege-1(imm070)*. Additionally, fully rescue (100%) of the poor PA14 survival phenotype in the *rege-1(imm070)* was observed upon abolishing the TORC1 signaling pathway, suggesting that the main factor leading to decreased survival in *rege-1(imm070)* is likely due to excess of TORC1 signaling. Furthermore, TORC1 signaling activated by *rege-1(imm070)* occurs before TORC1 is recruited to the lysosome.

To confirm the activation of TORC1 signaling in *rege-1(imm070)*, we investigated whether autophagosome formation was inhibited due to TORC1 signaling activation. We introduced the transgene GFP::LGG-1 into both N2 and *rege-1(imm070)* strains and blocked autophagy flux further by knocking down *rab-7*, a GTPase that is critical for autophagosome-lysosome fusion [38]. We then measured the autophagosome puncta of intestinal cells and observed a

significant decrease in GFP puncta in *rege-1(imm070)*. Our data, combined with previous findings, suggest that the decrease in autophagosome formation is likely due to the activation of TORC1 signaling in *rege-1(imm070)* (Fig 4C).

## Predicted ETS-4-targeted genes contribute to poor survival of PA14-fed *rege-1(imm070)*

To continue our investigation into the cause of TORC1 signaling activation, we shifted our focus to genes predicted to be targets of ETS-4. We identified potential ETS-4 targets using both previous motif search results and the ETS-4 ChIP data (S3 Table, [24,35]). Although both studies were performed in a wild-type background and did not show significant nuclear localization of ETS-4 like in defective *rege-1* reported previously, they provided a useful list of candidate genes for our initial investigation [24]. ETS-4 has been demonstrated to bind to a 5'-GGAA/T-3' core sequence through its ETS domain. A motif search further revealed 54 putative genes targeted by ETS-4, with five of them overlapping with *rege-1* negatively regulated genes, namely *acox-1.5*, *asp-12*, *clec-52*, *ech-9*, and *spp-3*. The ETS-4 ChIP analysis identified a total of 24 genes overlapped with *rege-1* negatively regulated genes (S3 Table), of which three overlapped with the five genes identified earlier, namely *asp-12*, *ech-9*, and *spp-3*. Although *asp-12*, a lysosomal aspartic-type endopeptidase, was even found to be down-regulated in the *rsks-1;daf-2* mutant in previous microarray data, all of these genes are class I genes (S5 Table). In contrast, *acox-1.5* (acyl-CoA oxidase 1, ACOX1) is the only class II gene identified through motif search. Although *acox-1.5* was not identified as an ETS-4 target by peak calling analyzed by ENCODE, two duplicates in the ETS-4 ChIP results showed a distinct peak at the promoter region around the *acox-1.5* transcription start site (S4A Fig). Acyl-CoA oxidase 1 (ACOX1) has several paralogs in *C. elegans*, one of which is *acox-1.5*. This enzyme plays a crucial role in the initial stage of the fatty acid β-oxidation pathway within the peroxisome, generating acetyl-CoA and other key intermediates required for cellular energy metabolism. Raptor acetylation is known to be essential for TORC1 signaling activation, and the cellular acetyl-CoA level is the key determinate. In addition, liver-specific ACOX1 knockout mice have shown a decrease in TORC1 signaling due to insufficient of acetyl-CoA [10]. Despite not being initially discovered through an ETS-4 motif search, we have included *ins-7* in our further investigation. This decision was based on our findings that *ins-7* transcription is stimulated in *rege-1(imm070)* and that ETS-4 ChIP analysis has identified it as one of targeted genes. Therefore, we investigated whether the poor PA14 survival phenotype in *rege-1(imm070)* could be rescued by either knocking down three genes (*asp-12*, *ech-9* from the class I list, and *acox-1.5* from class II list) using RNAi or incorporating the *ins-7(tm2001)* mutant.

To confirm that all the candidate genes are indeed negatively regulated by *rege-1*, we used quantitative PCR to measure their mRNA levels (Figs 3F and 5A). Consistent with our mRNA-seq results, the mRNA levels of *asp-12*, *ech-9*, and *acox-1.5* were up-regulated in *rege-1(imm070)* under both OP50- and PA14-fed conditions. Furthermore, the up-regulation of their mRNAs was abolished in either *ets-4(ok165)* or *rege-1(imm070); ets-4(ok165)* mutant (Fig 5A).

We next conducted a PA14 survival assay by incorporating *ins-7(tm2001)* with *rege-1(imm070)*. *Ins-7(tm2001)* harbors a deletion in the splicing donor of exon 2, leading to the complete loss of this exon. Notably, exon 2 is the largest exon present in *ins-7*. Although we did not observe significant increases in PA14 survival in *ins-7(tm2001)* alone, the combination of *ins-7(tm2001)* with *rege-1(imm070)* resulted in a 39% rescue in PA14 survival compared to *rege-1(imm070)* alone. (see Fig 5B and S1 Table).

*Ech-9* is one of the Enoyl-CoA Hydratase (ECH) paralog proteins in *C. elegans*. ECH is also an enzyme in the β-oxidation pathway in the peroxisome. The *C. elegans* genome contains six

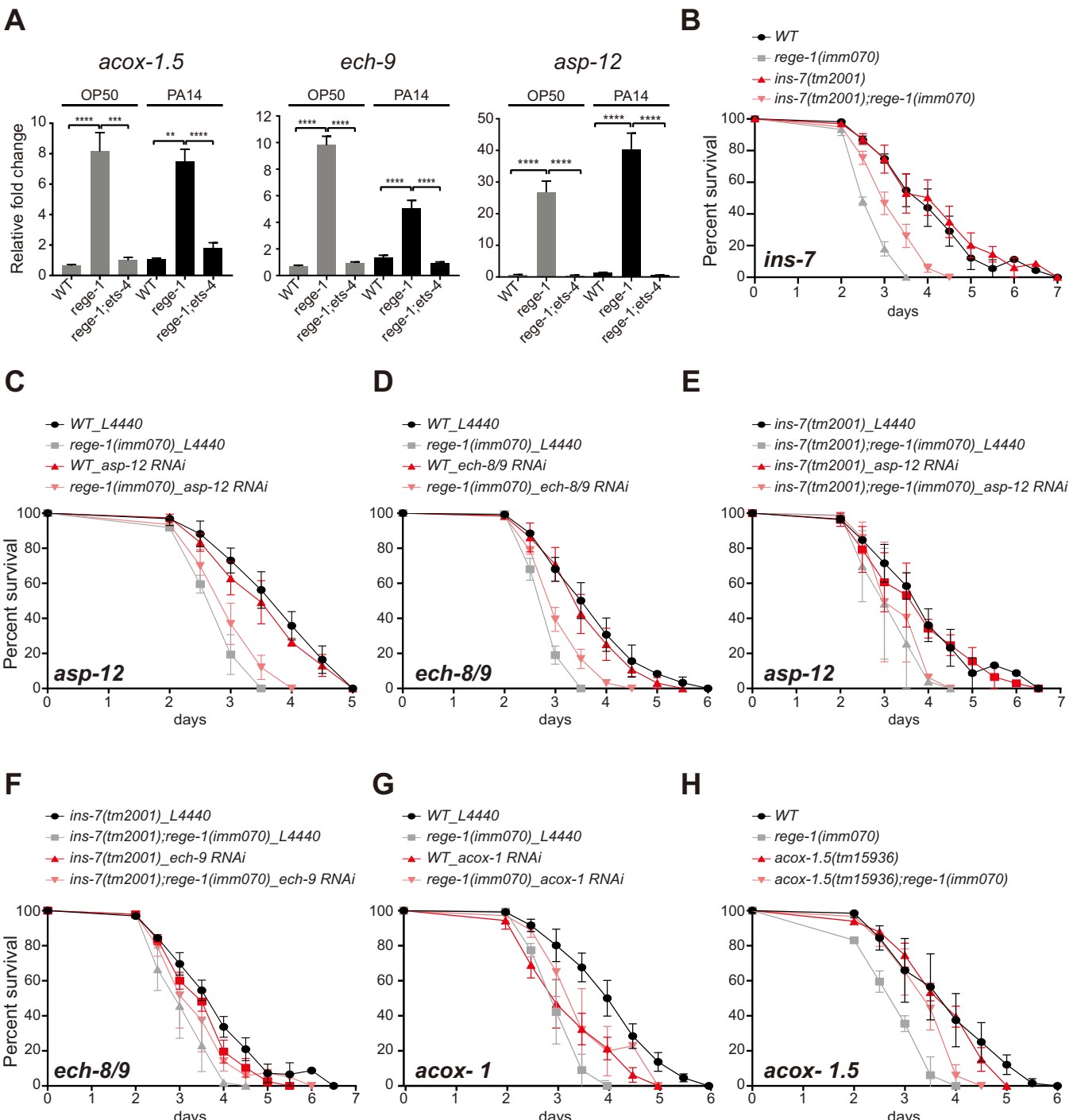

**Fig 5. Contribution of ETS-4 targeted genes to poor PA14 survival in *rege-1(imm070)*.** (A) Quantification of mRNAs of *asp-12*, *ech-9* and *acox-1.5* in wild-type, *rege-1(imm070)*, *ets-4(ok165)*, and *rege-1(imm070);ets-4(ok165)* fed with OP50 or PA14. Three biological replicates were performed for each sample. Statistical analysis was performed using one-way ANOVA with Tukey's multiple comparison test. (B) PA14 survival curves of wild-type, *rege-1(imm070)* with *ins-7(tm2001)* and *rege-1(imm070)*; *ins-7(tm2001)*. (C,D,G) PA14 survival curves of N2, *rege-1(imm070)* knockdown with (C) *asp-12*; (D) *ech-8/9*, (G) *acox-1*. (E,F) PA14 survival curves of *ins-7(tm2001)* and *rege-1(imm070)*; *ins-7(tm2001)* knockdown with (E) *asp-12*; (F) *ech-8/9*. L4440 was used as knockdown control (H) PA14 survival curves for wild-type, *rege-1(imm070)*, *acox-1.5(tm15936)*, and *rege-1(imm070); acox-1.5(tm15936)* are shown. Three biological replicates were performed for each survival curve. Raw data, p-values, and mean lifespans are presented in S1 Table.

paralogs of ACOX1 (*acox-1.1* to *acox-1.6*) and nine of ECHs (*ech-1* to *ech-9*) (S4B Fig). Among the nine ECHs, *ech-8* and *ech-9*, which have 69% protein identity, are predicted to be exclusively located in the peroxisome, while *ech-4*, which shares less than 20% identity with either *ech-8* or *ech-9*, is predicted to be located in both the mitochondria and peroxisome. Although all six peroxisomal ACOX1 proteins share high protein identity (ranging from 44% to 85%), *acox-1.5* is the least similar to the other ACOX1 proteins (ranging from 44 to 48%). Due to their high similarity in protein sequences, we next checked for potential off-target effects when using RNAi to knock down either *ech-9* or *acox-1.5*. In addition to targeting the *ech-9* mRNA, the *ech-9* RNAi sequence has a 58% sequence identity to *ech-8* mRNA, but not to *ech-4* mRNA. The RNAi clone generated for *acox-1.5*, which covers the entire cDNA sequence of *acox-1.5* (1980 bp), has 51% overlap with *acox-1.6*, 20–27% overlap with *acox-1.1* to *-1.3*, and 9% overlap with *acox-1.4*. Thus, the RNAi clone used for *ech-9* is likely to knock down both *ech-8* and *ech-9*, and the RNAi clone used for *acox-1.5* is likely to knock down other members of the ACOX1 family.

After knocking down either class I gene, *asp-12* and *ech-8/9*, we tested the PA14 survival of *rege-1(imm070)* and found no statistically significant difference in survival with or without *asp-12* RNAi (Fig 5C). However, knocking down *ech-8/9* in *rege-1(imm070)* resulted in a 36% increase in mean lifespan (Fig 5D and S1 Table). To enhance the rescue of *rege-1(imm070)* on PA14 survival, we carried out further PA14 survival assays using RNAi targeting either *asp-12* or *ech-8/9* on both *ins-7(tm2001)* and *ins-7(tm2001);rege-1(imm070)*. Although no further rescue was observed when knocking down *asp-12* in *ins-7(tm2001);rege-1(imm070)*, a rescue of approximately 69% relative to *rege-1(imm070)* was found when knocking down *ech-8/9* in *ins-7(tm2001);rege-1(imm070)* (Fig 5E and 5F)

While RNAi knockdown of *asp-12* or *ech-8/9* did not result in significant changes in PA14 survival, we found that worms with knocked down entire *acox-1* using *acox-1.5* RNAi had a mean lifespan of only 76% of wild-type on PA14, which is comparable to the survival reduction seen in *rege-1(imm070)* on PA14. However, when *acox-1* was knocked down in *rege-1(imm070)* worms, instead of observing further decreases in mean lifespan relative to either *rege-1(imm070)* or *acox-1* knockdown, a 10% lifespan extension was observed when compared to *acox-1* knocked down in wild-type worms. This suggests that knocking down the entire *acox-1* family contribute to some rescue of the poor lifespan of *rege-1(imm070)* worms on PA14, which is likely associated with an over-activated TORC1 signaling pathway (Fig 5G).

Since *acox-1.5* is the only acox-1 family member that is negatively regulated by *rege-1*, we performed the PA14 survival test using *acox-1.5(tm15936)*. This strain contains a 126 bp deletion spanning the 3' end of the first exon containing the splicing donor site, and only preserves the first 8 nucleotides. Fortunately, compared to wild-type worms, *acox-1.5(tm15936)* did not result in a shorter PA14 survival as *acox-1* knockdown. This finding partly supports our hypothesis that the shorter mean lifespan observed after knocking down *acox-1* is due to knocking down of entire acox-1 family genes. Excitingly, we found a significant 60.6% rescue of mean PA14 survival in *rege-1(imm070); acox-1.5(tm15936)* (Fig 5H). While not a complete rescue, the *acox-1.5(tm15936)* resulted in a substantial 60.6% rescue of mean PA14 survival in *rege-1(imm070)*, making it the most potent rescue achieved by deleting a single ETS-4 targeted gene.

## Mis-regulated genes in the IIS pathway are the primary contributors to the fat loss phenotype observed in *rege-1(imm070)*

Fat loss is the most prominent phenotype in *rege-1(imm070)*. We then determined the fat content using oil red staining to establish the relationship between fat loss and the poor PA14

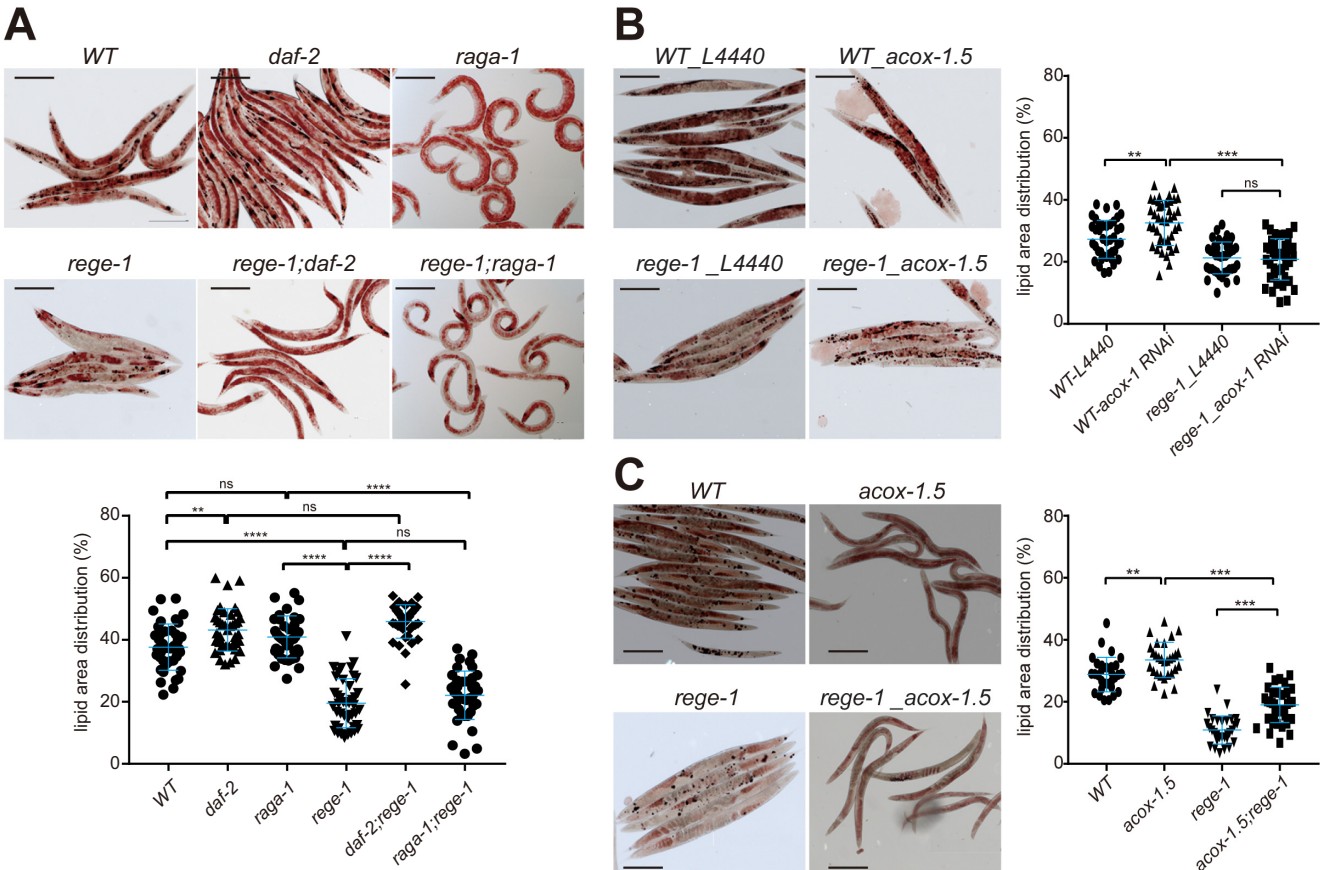

**Fig 6. The fat content changes in IIS pathway, TORC1 pathway and *acox-1.5*.** (A,C) Oil red staining was performed on wild-type, *rege-1(imm070)*, and mutant strains, including (A) *daf-2(e1370)* and *raga-1(ok386)*, and (C) *acox-1.5(tm15936)*, as well as their genetic combination with *rege-1(imm070)*. (B) Oil red staining was performed on wild-type and *rege-1(imm070)* worms treated with *acox-1.5* dsRNA knockdown, using L4440 was control. The quantification of staining is shown on the bottom(A) and right(B,C). The detailed quantification method is described in the Materials and Methods section. Briefly, the proportion of red area within a worm was calculated as the lipid-distributed area, and statistical significance was calculated using one-way ANOVA with Tukey's multiple comparison. Each experiment was conducted with 10–15 worms, and three independent replicates were performed for each experiment. The scale bars in the corner of the image represent 50μm (* $p < 0.05$, ** $p < 0.001$, *** $p < 0.0002$, **** $p < 0.0001$).

survival phenotypes of *rege-1(imm070)*. As *rege-1*-mediated short lifespan on PA14 can be rescued by abolishing IIS, TORC1 signaling, and partially rescued by the up-regulated genes in peroxisome fatty acid β-oxidation pathways, we examined the fat content in *rege-1(imm070)* in combination with either *daf-2(e1370)*, *raga-1(ok386)*, *acox-1.5(tm15936)* or *acox-1* knockdown (Fig 6A–6C). Previous studies have demonstrated that *daf-2* mutants have an increased fat content, which is consistent with our observations. Oil red staining revealed that *daf-2(e1370)* exhibited a significant increase in fat content, while *rege-1(imm070)* showed a significant decrease compared to wild-type (Fig 6A, [3,18]). Strikingly, no statistically significant difference was observed between *daf-2(e1370)* and *rege-1(imm070); daf-2(e1370)*, indicating that the suppression of the IIS pathway restores the fat loss phenotype (Fig 6A).

The role of TOR in lipid metabolism is multifaceted and not yet fully understood. Several studies have suggested that TOR activation promotes the accumulation of lipid droplets, while its inhibition leads to the breakdown of these droplets. In addition, TOR activation is known to inhibit autophagy, which is responsible for the degradation of cellular components, including lipid droplets [7,36,39]. To investigate the relationship between inhibiting TORC1 and the regulation of fat droplets, and whether this is linked to the fat loss phenotype in *rege-1*

*(imm070)*, we examined the fat content of N2, *rege-1(imm070)*, *raga-1(ok386)*, and *rege-1 (imm070);raga-1(ok386)*. However, we did not observe any significant differences in the amount of oil-red O stained area between wild-type and *raga-1(ok386)*, or between *rege-1 (imm070)* and *raga-1(ok386);rege-1(imm070)*. Our findings suggest that suppressing the TORC1 signaling pathway does not noticeably affect the fat loss phenotype of *rege-1(imm070)*. (Fig 6A).

We next examined the oil-red staining in both acox-1 knockdown and *acox-1.5(tm15936)*. Despite the existence of six different *acox-1* ortholog genes in the peroxisome, we observed a significant increase in fat content in *acox-1.5(tm15936)* when compared to wild-type, and similar increases in fat content were found in acox-1 knockdown (Fig 6B and 6C). Moreover, we observed a marked rise in fat content in *rege-1(imm070);acox-1.5(tm15936)* relative to *rege-1 (imm070)*. However, a noticeable decrease in fat content was still evident between wild-type and *rege-1(imm070);acox-1.5(tm15936)*, suggesting that *acox-1.5* not only plays a crucial role in lipid catabolism, but the decreased fat content seen in *rege-1(imm070)* is partly due to the upregulation of *acox-1.5* expression (Fig 6C). In contrast, we did not observe any increase in fat content in *acox-1* knockdown, which may be due to decreased lipid biosynthesis (see Discussion for more details).

To summarize, while the reduced survival of PA14-fed *rege-1(imm070)* is caused by the misregulation of both IIS and TORC1 signaling pathways, the fat loss phenotype in *rege-1 (imm070)* is predominantly due to the misregulation of IIS pathway genes. This misregulation, leads to the activation of the β-oxidation pathway in the peroxisome.

## Discussion

### Model

Effective regulation of the TOR signaling pathway is vital for balancing cellular growth and stress response, allowing cells to appropriately respond to environmental changes without compromising their functionality or viability. This study focuses on the effects caused by ETS-4, a transcription factor that responds selectively to stress. The amount of ETS-4 present in cells is tightly controlled by REGE-1, while its nuclear localization is hindered by DAF-16. When ETS-4 is overexpressed, it promotes the transcription of several class II genes, such as *ins-7*, which encodes for insulin-like peptide, and *acox-1.5*, which is involved in the peroxisomal β-oxidation pathway. The increased *ins-7* levels resulting from ETS-4 overexpression cause partial survival decrease upon exposure to pathogens, likely due to the activation of the IIS pathway. Furthermore, the increase in *acox-1.5* enhances peroxisomal β-oxidation, one of the lipid catabolic pathway, leading to excess acetyl-CoA production that promotes TORC1 signaling through increased acetylation of Raptor. Additionally, the resulting increase in TORC1 signaling caused by ETS-4 overexpression contributes to decreased survival under pathogen stress (Fig 7).

### Role of ETS-4 response to environmental stress

The function of ETS-4 is not yet fully understood. Although it has been observed that ETS-4 responds to environmental stimuli such as exposure to PA14 and cold stress by being up-regulated or shuttled into the nucleus, respectively, mutations in ets-4 do not seem to affect survival during PA14 exposure and may even increase longevity and resistance to cold stress [23,24]. This raises the question of why ETS-4 exists in the first place. However, recent research has revealed that ETS-4 plays a crucial role in neuron regeneration following laser damage, indicating a possible beneficial function in living organisms [25]. Moreover, phosphorylation of ETS-4 has been found to promote targeted gene expression, but further research is needed to

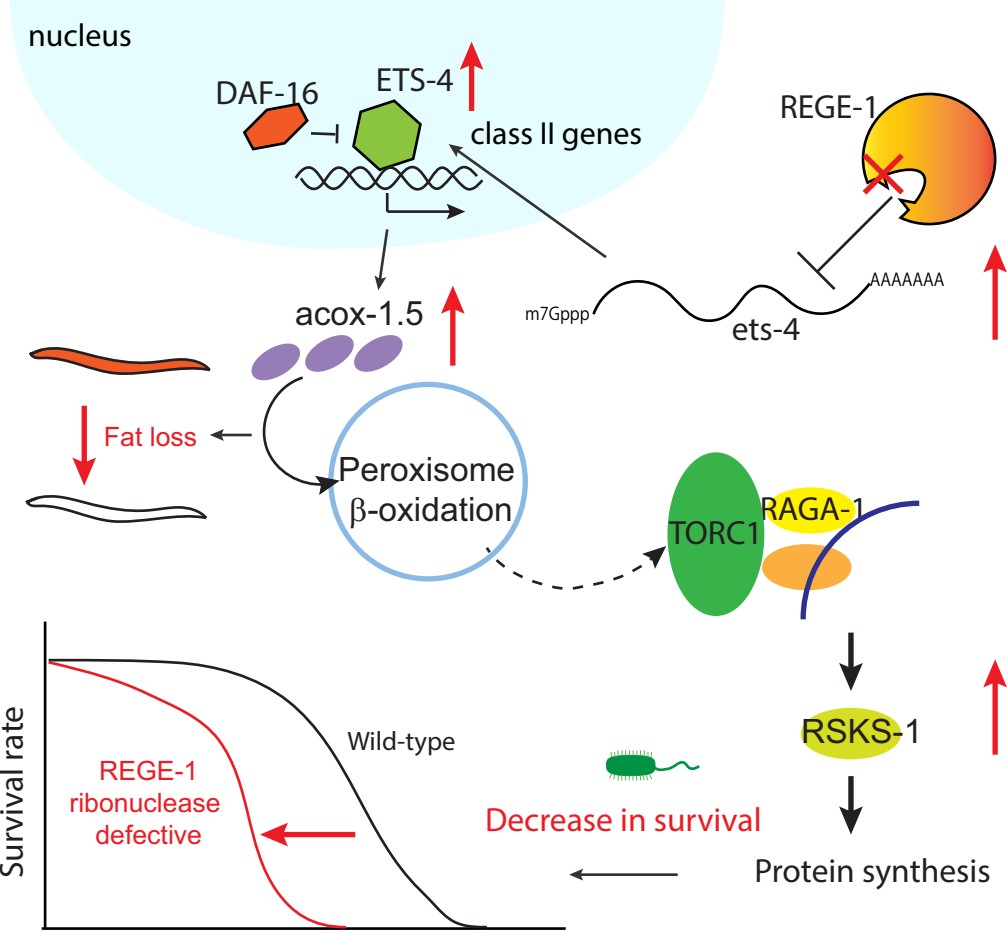

**Fig 7. The model of *rege-1* regulates lifespan and fat content in *C. elegans*.**

determine whether this modification affects ETS-4 nuclear localization and is essential for activating class II genes [25].

## Does the loss of fat contribute to decreased pathogen defense in *rege-1* mutants?

Fatty acids are essential for both basal and inducible immune responses, with γ-linolenic acid (GLA) and stearidonic acid (SDA) required for basal expression of innate immune effectors [40], and oleate needed for inducible immune responses [41]. However, the role of fatty acids in immune function is complex and not fully understood. While both *rege-1* single mutants and *raga-1;rege-1* double mutants exhibit a loss of fat phenotype, interestingly, only the *rege-1* mutant shows poor survival against PA14, while the *raga-1;rege-1* double mutant and wild-type have comparable survival rates. Although it is unclear if lipid composition is different between these mutants, it is unlikely that overall fat loss in the *rege-1* mutant is the sole cause of poor survival. In addition, though half of the fat loss in the *rege-1* mutant can be rescued by the *acox-1.5* mutant, a key enzyme in the peroxisomal β-oxidation pathway, and the *rege-1; acox-1.5* double mutation improves mean lifespan by 60% when exposed to PA14 might be interpreted as fatty acid contributed to survival though PA14, since all three unsaturated fatty acids involved in immunity are 18-carbon fatty acids, which is not believed to be

substrates of peroxisomal β-oxidation pathway [40–42]. And the *fat-3* mutant, which is defective in SDA synthesis, supplementing with a 20-carbon fatty acid, arachidonic acid, is insufficient to rescue the immune response, suggesting only specific kind of fatty acid could influence immune function. Thus, it is unlikely that an increase in the catabolic pathway of very long fatty acids in the *rege-1* mutant will result in changes in these three known immune-related fatty acid and further affect its immune function.

## The role of DAF-16, PQM-1 and ETS-4 in modulating longevity genes

The longevity contributed by the IIS pathway in *C. elegans* is known to be due to increased expression of class I genes and repression of class II genes. Although DAF-16 is known to promote class I gene expression, how class II genes are repressed remains largely unknown. A study by Tepper et al. (Cell, 2013) provided a good candidate, PQM-1, which binds to most of the promoters at class II genes. However, one puzzle is that deleting PQM-1, which is localized in nucleus most of the time, does not result in lifespan extension, even though class II gene repression during longevity in *daf-2* mutant can be explained by DAF-16 excluding PQM-1 from the nucleus [6]. On the other hand, ETS-4 acts as a transcription factor for class II genes, and its deletion results in extended lifespan, as shown by Thyagarajan et al. (PLOS Genetics, 2010, [24]). An interesting finding is that knocking down *akt-1/2*, which are key components that prevent DAF-16 from entering the nucleus, leads to an additive increase in lifespan when combined with an *ets-4* mutation. This observation supports the idea that class I genes are regulated by DAF-16, while class II genes are regulated by ETS-4. Although more DAF-16 is expected to shuttle into the nucleus in the *akt-1/2* RNAi knockdown background, it is possible that some ETS-4 may still remain in the nucleus. Therefore, it is reasonable to expect that the *ets-4;akt-1/2* RNAi mutant would have a longer lifespan than the *akt-1/2* RNAi alone.

It is important to recognize that ETS-4 is not the sole transcription factor involved in regulating class II genes, as PQM-1 is also involved. A study by Tepper et al. (Cell, 2013) has indicated that PQM-1 promotes the transcription of both class I and II genes [6]. It is possible that the lack of lifespan extension in the *pqm-1* mutant is due to reduced expression of both class I and II genes, leading to no significant changes in lifespan.

## *Rege-1* regulation of TORC1 signaling pathway is an evolutionary conserved process

*Rege-1*-mediated inhibition of TORC1 signaling is not restricted to *C. elegans*, as a previous study showed that mTOR signaling is activated in intestine-specific *Reganase-1* knockout mice [17]. While inhibition of mTOR in mouse intestine enhances proliferation after tissue damage and further results in better survival, genetic inhibition of TOR signaling can promote longevity and improve stress responses [43,44]. A recent study showed that neuron-specific expression of *raga-1* in the *raga-1* mutant is sufficient to abolish the mutant's longevity phenotype, suggesting that activation of TOR signaling in different tissues can cause different outcomes than activation in the whole animal [45]. Therefore, the issue of whether the poor survival of PA14-fed *rege-1(imm070)* is due to tissue-specific misregulation of IIS and TORC1 signaling pathways remains an open question.

## The impact of peroxisomal fatty acid β-oxidation on *rege-1* mutant phenotypes

Although we observed a 60% rescue of the poor PA14 survival phenotype in *rege-1* mutants with the *acox-1.5* mutation, there are still three unanswered questions. Firstly, knocking down

*acox-1.5* using RNAi resulted in similar survival in *rege-1* mutants and in the wild-type, so why was there only a 60% rescue in the *acox-1.5* mutation? Secondly, although the fat content was rescued in the *acox-1.5* mutant, why did knocking down *acox-1.5* using RNAi result in increase in fat as in the *acox-1.5* mutant, but no rescue of the fat loss phenotype was observed in knocking down *acox-1.5* in *rege-1* mutants? Finally, two peroxisomal fatty acid β-oxidation pathway genes, *acox-1.5* and *ech-9*, were found to be up-regulated genes in the *rege-1* mutant, so why were there different degrees of rescue in each case?

Possible explanations for these puzzles are as follows: Although six similar acyl-coA oxidase 1 (*acox-1*) paralogs (40% above identity) are found in *C. elegans*, we still observed fat accumulation in the *acox-1.5* mutant alone, suggesting that the functions of these *acox-1* paralogs are distinct. As outlined in the Results section, the dsRNA produced from the *acox-1.5* coding sequence can be used to target other *acox-1* genes, effectively inhibiting the entire peroxisomal β-oxidation pathway. While the *acox-1.5* mutant alone can rescue the increase in β-oxidation resulting from heightened *acox-1.5* expression, the augmented β-oxidation pathway in the *rege-1* mutant caused by elevated *ech-9* can only be rescued by knocking down all *acox-1* paralogs, not just the *acox-1.5* mutant alone. Therefore, coalescing both *acox-1.5* and *ech-9* mutants in the *rege-1* mutant could potentially lead to complete rescue of PA14 survival in the *rege-1* mutant. On the other hand, rather than observing a rescue of fat loss as seen in *acox-1.5* mutants, we observed a significant decrease in fat when knocking down *acox-1* in the *rege-1* mutant. Our previous observations suggest that the fully PA14 survival rescue is likely due to the inhibition of the TORC1 signaling pathway, and the PA14 survival is similar in *acox-1* knocking down in *rege-1* and in wild-type. Therefore, we would not expect the strong decrease in fat to be due to further breakdown of fatty acids that accumulated after knocking down *acox-1*, since that would generate more acetyl-CoA to promote TORC1 signaling. One possible explanation is that the DEGs in the *rege-1* mutant, likely the increase of *ech-9*, have feedback mechanisms to prevent further fatty acid synthesis. This might explain why we observed a stronger reduction in fatty acids when knocking down *acox-1*s in the *rege-1* mutant. The third question is why knocking down *ech-9* only resulted in a 36% rescue rate, whereas knocking down *acox-1*s resulted in no difference between rege-1 and wild-type. One possible explanation is that there is a peroxisomal located *ech-4* gene that will not be targeted by dsRNA fragments generated from *ech-9*. Thus, even if the increase in *ech-9* in the *rege-1* mutant can be alleviated by knocking down *ech-9*, it does not fully prevent the excess of *acox-1.5* in *rege-1* mutant, and further producing more acetyl-CoA. This results in only a 30% rescue rate when knocking down *ech-9* in the *rege-1* mutant.

## Materials and methods

### Genetics

*C. elegans* maintenance and genetics were performed as described elsewhere [46]. The wild-type strain used in this study was Bristol N2. Alleles used in this study are listed in the order of chromosome, LG I: *rege-1(imm070)*, *rege-1(tm2265)*; LG II: *pqm-1(tm8184)*, *raga-1(ok386)*; LG III: *acox-1.5(tm15936)*, *daf-2(e1730)*, *rsks-1(tm1714)*; LG IV: *ins-7(tm2001)*; LG X: *ets-4 (ok165)*.

### Generation of *rege-1(imm070)* allele

CRISPR/Cas9 genome editing was performed in *C. elegans* as previously described with some modifications [47]. In brief, a 20μl CRISPR injection mixture containing 4.2μM tracrRNA, 2.5μM target sgRNA, 1.7μM *rol-6* sgRNA, 1.56μM recombinant Alt-RspCas9 protein (~5μg), and 2.5μM of each repair template for *rol-6* and the target gene was prepared. All reagents

were purchased from Integrated DNA Technologies, Inc. The tracrRNA, sgRNAs, and Cas9 protein were first incubated at 37˚C for 10 minutes in duplex buffer (Integrated DNA Technologies, Inc.) for reconstitution, followed by mixing with the repair templates. Prior to injection, the mixture was filtered through a 0.22μm cellulose filter (Corning Life Sciences). Twenty to thirty young adult worms were injected and maintained individually on plates. The F1 rollers were further maintained and their genotypes were checked. The sequences for the sgRNA and repair templates are listed in (S6 Table).

## *P. aeruginosa* (PA14) killing assays

The PA14 standard slow killing assay (SKA) was performed as previously described [48]. Briefly, 5 μL of PA14 overnight culture was seeded in the center of SKA plates (3.5-cm agar plates containing NGM with 0.35% instead of 0.25% bactopeptone). The plates were incubated at 37˚C for 24 h and then kept at 25˚C before use. Embryos were obtained by hypochlorite treatment of adult worms using 0.25 μM NaOH and 20% bleach, and were plated and grown on OP50 at 20˚C until they reached the L4 stage. A total of 70 to 100 L4 worms were then transferred to PA14-seeded plates (~35 worms per plate) and maintained at 25˚C. All synchronization and maintenance were identical unless noted otherwise. The worms were transferred to new plates every 1–2 days and were scored twice a day. Death was recorded if a worm did not respond to prodding. OASIS (http://sbi.postech.ac.kr/oasis) was used to perform statistical analysis of survival data, and p-values were calculated with a log-rank (Mantel-Cox method) test [49]. Experiments involving targeted gene knockdown by feeding RNAi were performed by growing wild-type and *rege-1(imm070)* worms on NGM plates with 1 mM IPTG (Isopropyl β-D-1-thiogalactopyranoside) seeded with HT115 bacteria containing fragments of targeted genes. The bacteria were either from the Ahringer C. elegans RNAi feeding library or the cDNA was cloned into the L4440 vector.

## mRNAseq and analysis

Total RNA was extracted from groups of 100 two-day-old adult worms grown at 25˚C using TRIzol reagent (Invitrogen) for both RT-qPCR and mRNAseq preparation. The synchronization and growth conditions were the same as previously described. Briefly, embryos were obtained after hypochlorite treatment of adult worms and were then plated on OP50 at 20˚C until the L4 stage. The worms were then transferred onto either PA14- or OP50-seeded plates at 25˚C for 2 days before being picked for RNA extraction. Libraries were constructed using the KAPA mRNA HyperPrep Kit (Roche) and KAPA Dual-Indexed Adapter Kit (Roche) for Illumina Platforms. Sequencing was performed by Genomics BioSci & Tech Inc. (New Taipei City, Taiwan). The sequencing data were uploaded to the Galaxy web platform, and the public server at usegalaxy.org was used for data analysis [50]. Sequencing reads were aligned to the C. elegans genome WS245 (ce11) using HISAT2 (v 2.2.1) [51]. Aligned transcripts were quantified with featureCounts (v 2.0.1) [52], followed by DESeq2 (v 2.11.40.7) [53]to obtain differentially expressed genes (adjusted p-value < 0.05) and Kallisto quant (version 0.46.2) to obtain the abundances of RNA-Seq transcripts (transcripts per million, TPM) [38,54]. All of the analysis processes mentioned above were performed on Galaxy [50]. The heatmaps of ETS-4 regulated genes were plotted using Heatmapper [55] (http://www.heatmapper.ca/expression/). To determine the significance of the overlap between two groups of genes, we performed a one-way ANOVA analysis, using the number of overlapping genes as the basis for calculating whether the differences between the random selection and experimental dataset were statistically significant. The raw data for mRNAseq can be found in Gene Expression Omnibus with accession number GSE218235.

## ENCODE ChIP-seq data

The differentially expressed genes (DEGs) of *rege-1* mutants that were either fed with OP50 or PA14 were used to extract the ENCODE ChIP-seq data associated with DAF-16 (ENCFF681MIE), PQM-1 (ENCFF068DOO), and ETS-4 (ENCFF938QXU). The ChIP-seq read intensities were extracted from 1.5K base pairs upstream and downstream of the transcription start site (TSS) for each DEG and were visualized in heatmap format. To compare the ChIP-seq intensity distributions among different groups, the intensity of the reads at all 3K positions relative to the TSS of the same group of DEGs was averaged. To determine whether the promoter of a given gene was bound by ETS-4, the peak calling file (ENCFF037IAO) was utilized, and any called peak that overlapped 1K base pairs upstream or downstream from the TSS was considered as a promoter region bound by ETS-4.

## RT-qPCR

About 0.3 μg of RNA was used to prepare cDNA with M-MuLV reverse transcriptase (Protech). Relative quantification was measured by the standard curve method. In brief, serial 5-fold dilutions of cDNA were used as standard curves. One microliter of 25-fold-diluted cDNA and 0.4 μM primer was used with 2x qPCRBIO SyGreen Blue Mix. Quantitative PCR was performed and analyzed using StepOnePlus (Applied biosystem). *Act-3* mRNA was used as the internal control. Statistical analysis on all experiments was performed using Prism 7 (GraphPad). The statistical methods used to calculate p-values are indicated in the figure legends. The primers used in this study are listed in (S6 Table).

## PA14 clearance and avoidance assay

The PA14 clearance assay involved transferring L4 stage worms fed with OP50 to plates seeded with PA14-RFP lawn for 24 hours, followed by transferring them to an OP50-seeded plate for an additional day. Images were taken before and after the worms were transferred to OP50 after PA14-RFP exposure using a 10× Zeiss objective on a Zeiss fluorescence microscope (Axio Imager.M2) with Axiovision 4.8 software in both the red and transmitted light channels (with differential interference contrast). Briefly, the area of the worms was defined using the ROI tool of ImageJ (NIH; http://rsb.info.nih.gov/ij/) from the DIC image. The same region from the DIC images was used to define the area of fluorescence regions. Each experiment was scored with 10–15 worms, and three independent repeats were performed. The quantitative data were displayed using Prism 7. The PA14 avoidance assay was performed by plating L4 stage worms fed with OP50 onto PA14-seeded plates overnight and scoring the number of worms either on or off the PA14 lawn.

## Fluorescent imaging

*Pins-7::gfp* **expression.** Approximately thirty L4 worms grown on OP50, containing the *pins-7::gfp* transgene in either wild-type or *rege-1(imm070)* backgrounds, were transferred to new OP50 plates and grown at 25˚C for two days. On the second day, adult *C. elegans* were anesthetized in 10 mM tetramisole on 2% agar pads. *Pins-7::gfp* images were acquired in the green and transmitted light channels (with differential interference contrast, DIC) using a 10× Zeiss objective on a Zeiss fluorescence microscope (Axio Imager.M2) with Axiovision 4.8 software. The fluorescence intensity of the animals was quantified with ImageJ (NIH; http://rsb.info.nih.gov/ij/). Briefly, the area of the worms was defined by the ROI tool of ImageJ from the DIC image. The same region from DIC images was used to define the area of the fluorescence

regions. Each experiment was scored with 10–15 worms, and three independent repeats were performed for each experiment.

**Autophagosome formation using GFP::LGG-1.** Semi-synchronized embryos from either wild-type or *rege-1(imm070)* adult worms containing *gfp::lgg-1* were obtained after killing the adults with hypochlorite. The embryos were grown on L4440 or *rab-7* RNAi clone 1mM IPTG containing plates (obtained from the Ahringer *C. elegans* RNAi feeding library) at 20˚C until most of them reached the L4 stage. Twenty to thirty worms were then transferred to new RNAi plates with the same RNAi food and grown at 25˚C for an additional two days before imaging. The adult *C. elegans* were anesthetized with 10 mM tetramisole on 2% agar pads, and confocal images were acquired using a Zeiss LSM700 confocal microscope. The GFP::LGG-1 puncta were quantified using ImageJ (NIH; http://rsb.info.nih.gov/ij/). In brief, the two intestinal cells proximal to the terminal bulb of the pharynx of every worm imaged were defined as the region of interest. The number of puncta was calculated using binary tools (make binary and watershed), followed by analyzing particle numbers by setting the particle threshold to 50 and quantifying particle numbers. The particle counts were further normalized to the area of the cell. The statistical significance between two samples was calculated using Prism with an unpaired T test.

**Quantification of nuclear intensity of *ets-4:gfp* transgene.** To determine the localization of ETS-4, we constructed a plasmid (*pets-4::ets-4::gfp::unc-54*) containing the *ets-4* promoter (2149bp), coding region (1389bp), *gfp* sequence, and *unc-54* 3'UTR, which we co-injected with a *rol-6* dominant mutant plasmid (pRF4) to generate transgenic strains. Using a Zeiss fluorescence microscope (Axio Imager.M2) and Axiovision 4.8 software, we acquired images in the green and transmitted light channels (with differential interference contrast) using a 10x objective. We defined the nucleus and intestine cell regions of worms using the ROI tool in ImageJ (NIH; http://rsb.info.nih.gov/ij/) from DIC and GFP images. The ratio of nuclear intensity was calculated as the intensity of the nucleus divided by the intensity of the intestine cell. The ratio of all strains injected with the *ets-4:gfp* plasmid was calculated relative to the N2 worms that did not receive the plasmid. We quantified 2–3 nuclei in each of 3–5 worms with higher ETS-4 expression due to inconsistent cell-to-cell localization. Prism 7 was used for statistical analysis.

## Oil red O staining and image quantification

Oil red O staining and image quantification were performed as previously described [22,56]. Briefly, 500–1000 OP50-fed day-1 adult animals were fixed in 2% paraformaldehyde (PFA) in MRWB buffer (160 mM KCl, 40 mM NaCl, 14 mM Na2EGTA, 1 mM spermidine-HCl, 0.4 mM spermine, 30 mM Na-PIPES pH 7.4, 0.2% β-mercaptoethanol) for 1 hour at room temperature. Samples were then incubated in 60% isopropanol for 15 minutes at room temperature, followed by overnight staining in 60% Oil-Red-O on a rocking platform. Animals were mounted and imaged using a Zeiss M1 color camera outfitted with DIC optics. The oil red intensity of the animals was quantified using ImageJ. Briefly, after background subtraction, individual worms were selected as regions of interest. The color threshold was used to select red pixels, and the area of red pixels relative to the whole area of the worm was calculated as the lipid-distributed area.

## Supporting information

**S1 Fig. Wild-type and rege-1(imm070) exhibit similar pathogen clearance and avoidance abilities.** (A-B) RFP-labeled PA14 levels were measured in the intestinal lumen of worms. (A) Images were taken either prior to (day 1) or after the worms were transferred to OP50 after

PA14-RFP exposure (day 2). The same region from the DIC images was used to define the area of fluorescence regions. (B) The quantification of PA14-RFP intensity was performed on 10–15 worms per experiment. (C) Lawn occupancy of animals on PA14 was evaluated after 24 hours in two different rege-1 strains, *rege-1 (tm2265)* (left) and rege-1 ribonuclease defective strain *rege-1(imm070)* (right), in combination with *ets-4(ok165)*. Thirty L4 stage worms were transferred to the center of the PA14 lawn, at 25˚C, and scored for avoidance after 24 hours (EPS)

**S2 Fig. Statistic analyses of the differential expressed genes (DEGs) in rege-1(imm070).** (A) The Venn diagram shows the overlapping DEGs between rege-1*(imm070)* relative to wild-type or *rege-1(imm070); ets-4(ok165)* for both OP50- and PA14-fed samples. (B) Principal component analysis was performed on all mRNAseq samples. Each sample was replicated two to three times. Samples with the same genetic background are labeled and circled in the same color. WT: N2, rege-1: *rege-1(imm070)*, rege-1;ets-4:*rege-1(imm070);ets-4(ok165)*. (C)Volcano plots show *rege-1(imm070)* versus either N2 or *rege-1(imm070);ets-4(ok165)* fed with OP50 or PA14. The x-axis shows the magnitude of fold change (FC) in $\log_2$ value, and the y-axis shows the statistical significance (P value) in $-\log_{10}$ value. All dots in the figure represent genes that shown statistically significant changes (p<0.05). Red and blue dots indicate genes are up- and down-regulated in *rege-1(imm070)*, respectively. Genes that mentioned in the main text are labeled. (D) Enrichment analyses were performed for tissue expression patterns (TEA), pheno-types expression patterns (PEA), and gene ontology (GO) in DEGs that are up-regulated in *rege-1(imm070)* relative to both wild-type and *rege-1(imm070);ets-4(ok165)* in OP50- (OP_up) and PA14-(PA_up) fed samples or down-regulated in *rege-1(imm070)* relative to both wild-type and *rege-1(imm070);ets-4(ok165)* in OP50- (OP_down) and PA14-(PA_down) fed samples. OP and PA denote OP50- and PA14- fed samples. DN: *rege-1(imm070)*, WT: N2 or double: *rege-1(imm070); ets-4(ok165)* (EPS)

**S3 Fig. The statistical analysis of the correlation between various ChIP data and groups of mRNAs.** (A) Statistical significance of Class 2 genes, ChIP data from DAF-16, PQM-1 and ETS-4 in published ETS-4 positively regulated genes. (B)ETS-4 ChIP data in published DEGs of *daf-2;rsks-1* that overlapped with rege-1 negatively regulated genes (C) gene expression profiles from mRNAseq of all the *C. elegans* orthologs found in the TOR pathway, core components, interactors, substrates, or regulators. Orange: OP50 fed, Green: PA14 fed. (EPS)

**S4 Fig. Related to Fig 5.** (A) The genome browser shows ENCODE ETS-4 CHIP data around *acox-1.5*. (B) protein identity comparison between *ACOX-1.1–1.6* (left) and peroxisome enoyl-CoA hydratase (ECH-4, ECH-8 and *ECH-9*)(right). All numbers are shown in percentage of identity. (C) Quantification of clec-52 and spp-3 mRNA changes in wild-type, *rege-1 (imm070)*, *ets-4(ok165)* and *rege-1(imm070);ets-4(ok165)*. (EPS)

**S1 Table. Summary of lifespan analysis statistic of survival curve.** (XLSX)

**S2 Table. Log(2) Fold Change Values of Differentially Expressed Genes in rege-1(imm070) y.** (XLSX)

**S3 Table. Genes identified in Encode ETS-4 CHIP peaks.** (XLSX)

**S4 Table. Comparison of Differentially Expressed Genes in This Study and Previously Published Datasets.**
(DOCX)

**S5 Table. The gene list of *daf-2(e1370);rsks-1(ok1255)* used for pearson r correlation with the *rege-1(imm070)* differential expressed genes fed with OP50 or PA14.**
(XLSX)

**S6 Table. primer used in this study.**
(DOCX)

## Acknowledgments

We thank The *C. elegans* Core Facility of the National Core Facility for Biopharmaceuticals, Ministry of Science and Technology, Taiwan, for helping with various worm-related equipment and reagents. *Rege-1*(tm2265), *rsks-1(tm1714)*, *pqm-1(tm8184)*, *acox-1.5(tm15936)* was provided by National Bio-Resource Project (NBRP), Tokyo Women's Medical University, School of Medicine, Tokyo, Japan. *daf-2(e1370)*, *ins-7(tm2001)*, *raga-1(ok386)* and *ets-4 (ok165)* were provided by the *Caenorhabditis* Genetics Center, which is funded by NIH Office of Research Infrastructure Programs (P40OD010440). PA14 harboring RFP are kindly gifted from Dr. Chang-Shi Chen, Department of Biochemistry and Molecular Biology, National Cheng Kung University, Tainan City, Taiwan. We also thank the imaging core at the First Core Labs, National Taiwan University College of Medicine, for the technical support in image acquisition and analysis. We acknowledge the ENCODE consortium and the ENCODE production laboratories, Kevin White, UChicago, Michael Snyder, Stanford, generating the ETS-4, DAF-16 and PQM-1 datasets. We also grateful for the Galaxy server that was used for some calculations is in part funded by Collaborative Research Centre 992 Medical Epigenetics (DFG grant SFB 992/1 2012) and German Federal Ministry of Education and Research (BMBF grants 031 A538A/A538C RBC, 031L0101B/031L0101C de.NBI-epi, 031L0106 de.STAIR (de. NBI)).

## Author Contributions

**Conceptualization:** Hsin-Yue Tsai.

**Data curation:** Yi-Ting Tsai, Chen-Hsi Chang, Hsin-Yue Tsai.

**Formal analysis:** Yi-Ting Tsai, Chen-Hsi Chang.

**Funding acquisition:** Hsin-Yue Tsai.

**Methodology:** Yi-Ting Tsai, Hsin-Yue Tsai.

**Project administration:** Hsin-Yue Tsai.

**Supervision:** Hsin-Yue Tsai.

**Validation:** Yi-Ting Tsai.

**Writing – original draft:** Yi-Ting Tsai.

**Writing – review & editing:** Hsin-Yue Tsai.

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
