## [Decision Letter · Decision Letter 0]

6 Jan 2023

Dear Dr TSAI,

Thank you very much for submitting your Research Article entitled 'Rege-1 promotes C. elegans survival by modulating metabolic pathways' to PLOS Genetics.

The manuscript was fully evaluated at the editorial level and by independent peer reviewers. The reviewers appreciated the attention to an important problem, but raised some substantial concerns about the current manuscript. Based on the reviews, we will not be able to accept this version of the manuscript, but we would be willing to review a much-revised version. We cannot, of course, promise publication at that time.

If you decide to revise the manuscript for further consideration at PLOS Genetics, please aim to resubmit within the next 60 days, unless it will take extra time to address the concerns of the reviewers, in which case we would appreciate an expected resubmission date by email to plosgenetics@plos.org.

We are sorry that we cannot be more positive about your manuscript at this stage. Please do not hesitate to contact us if you have any concerns or questions.

Yours sincerely,

Coleen T. Murphy

Academic Editor

PLOS Genetics

Gregory P. Copenhaver

Editor-in-Chief

PLOS Genetics

Reviewer's Responses to Questions

**Comments to the Authors:**

Reviewer #1: My review has been uploaded as an attachment.

Reviewer #2: Regen-1 is an immunity gene and knockout in mice has divergent outcomes depending on whether the knockout is whole-body or tissue-specific. In this study, Tsai et al. use C. elegans with mutated rege-1 (the sole ortholog of mammalian Regen-1) to test its role in survival and immunity. The authors find that the rege-1 mutant has decreased lifespan at baseline and decreased survival following exposure to Pseudomonas aeruginosa. Mutation of the rege-1 ribonuclease activity site alone is sufficient to phenocopy this decreased survival.

rege-1 mutants have elevated levels of its known target mRNA, ets-4. RNA sequencing on rege-1 mutants and rege-1;ets-4 double mutants shows that ets-4-regulated genes are enriched for genes in the insulin/insulin-like signaling (IIS) and target of rapamycin complex I (TORC1) pathways. Crossing the rege-1 mutant to the daf-2 (IIS) or rsks-1 (TORC1) mutants improves survival and partially rescues the fat loss phenotype of the rege-1 mutant. Tsai et al. also test the effect of crossing the rege-1 mutant to ins-7 and acox-1.5 (targets of ets-4) mutants and saw mild improvements in survival. Overall, the authors conclude that in a rege-1 mutant, excess ets-4 upregulates IIS and TORC1 signaling, leading to poor survival. This study adds contributes to our knowledge of how an understudied gene, rege-1, is involved in survival and immunity and highlights the role of the nutrient metabolism pathways IIS and TORC1 in mediating rege-1¬ survival.

However currently there are some concerns over data analysis supporting the key findings, statisics and replicates that leave this work short of being publication ready.

Key Issues to be Improved:

• A key finding of the paper from the RNA seq is the changes in PQM related genes. More data is needed to flesh out this finding – is PQM localization changed v DAF-16? Does modulating PQM impact the effects of rege-1 or the daf-2 ‘rescue’?

• Effects of mTORC1: Is activity of mTORC1 altered in rege-1 mutants? This could be examined using phos S6K/RSKS-1 antibodies used elsewhere in C. elegans studies

• The authors should provide more details about mutants used in the study to assist with proper interpretation of the results. How much of the rege-1(tm2265) coding sequence is deleted? Is it a total null or is there functional protein remaining? Is the ribonuclease activity site that is mutated in rege-1(ht070) also mutated in the tm2265 allele?. Verify that the rege-1(ht070) mutation abolishes ribonuclease activity – noy citation is given and the fig cited does not carry out those exps. Give clarifying details about the ets-4(ok165) mutant. Is it a deletion? Where and how much of the gene is deleted? Same for the ins-7 mutant used.

• Some figures have missing statistics, replicates, or information about how many replicates were performed: No replicate information for the figures in S1. No statistics in S1D or S1E. S1B and S1C only have 1 replicate

• The authors should check that all experiments have clearly explained methods either in the Methods section and/or in the figure legend. For example, for figure S1E:

o No information on the methods of the imaging performed. For example how old were the worms and are they synchronized?

o The worms in the far right bottom panel of S1E look morphologically distinct from the ones above (which are supposedly the same genotype).

o Total RFP in the intestine is used as a proxy for bacterial clearance. How does total amount of RFP in intestine tell you anything about clearance? For this it would require an image while they feed, then remove them from the RFP-PA14 lawn for a set amount of time and then image again? Can the authors provide more details about how this experiment was performed in the Methods section?

• Although in figure 4 rege-1 loss indeed does not suppress lifespan in the presence of acox-1.5 RNAi the result is more complicated than reported in the text and needs further elaboration. There is an interesting contextual effect of acox-1.5 RNAi in that it shortens the lifespan of WT animals but increases the lifespan of rege-1 mutants. This shd be tested statistically and discussed in results.

Minor Points

• In fig 4C stats in WT v WT+ RNAi need to be included

• Line 64 – “Pseudomonas” spelled incorrectly

• Line 107 – An additional introductory sentence regarding what’s known about ets-4 would improve this section. What’s known about it? Is it a transcription factor? What genes is it known to regulate?

• Line 153 – How were the results from WormExp v2.0 verified and why do the numbers differ after verification? Add details in Methods section.

• Line 155 – When comparing the overlapping genes, the denominators are 41 and 17. Where do those numbers come from? They aren’t in the Venn Diagram. Many comparisons are being made from the same RNA sequencing experiment and clarifying how the genes are begin subset would help the reader better interpret the results.

• Line 186 – If the RNA sequencing changes were similar to the transcriptional profile of a rsks-1 mutant, why cross to a raga-1 mutant? Some justification for this choice in the text would clarify for the reader

• Line 269 - a more direct test of whether TOR pathway activity is increased would be to do Westerns for phosphorylation of TOR or its downstream targets

• Line 293 – needs citation

**Have all data underlying the figures and results presented in the manuscript been provided?**

Reviewer #1: **No: **The coversheet indicates that the RNAseq dataset has been deposited in GEO, but the authors do not mention this in the text, nor do they provide a link to the data. At a minimum, an Excel file with the complete list of differentially expressed genes along with their statistical significance should have been provided to reviewers.

Reviewer #2: Yes

PLOS authors have the option to publish the peer review history of their article (what does this mean?). If published, this will include your full peer review and any attached files.

Reviewer #1: No

Reviewer #2: No

---

## [Decision Letter · Decision Letter 1]

12 Jul 2023

Dear Dr TSAI,

We are pleased to inform you that your manuscript entitled "Rege-1 promotes C. elegans survival by modulating metabolic pathways" has been editorially accepted for publication in PLOS Genetics. Congratulations!

While the manuscript is in principle accepted, in the final draft that you prepare for the production team, please be sure to address all of the remaining points that the reviewers have raised, particularly regarding grammar and spelling issues and correctly identifying reagents (i.e. RNAi vector L4440 is frequently improperly referred to as L440).

Yours sincerely,

Coleen T. Murphy

Academic Editor

PLOS Genetics

Gregory P. Copenhaver

Editor-in-Chief

PLOS Genetics

Comments from the reviewers (if applicable):

Reviewer's Responses to Questions

**Comments to the Authors:**

Reviewer #1: My review is uploaded as an attachment.

Reviewer #2: The revision has added some new data on additional TORC1 pathway and ACOX1 mutants and some negative data on PQM. These new data support the role of TORC1 levels in the PA phenotype at least genetically but now make it less likely that PQM is involved - and the manuscript now has some discussion trying to explain this lack of an effect.

Overall the revision has fixed a lot of the previous errors and I believe this work is suitable for PLoS genetics as it adds some interesting new interaction data between key signaling pathways, pathogen resistance and REGE-1/ETS-4. Although mechanistic detail is lacking the revision has added some clearer mutant analyses and epistasis and I recommend publication.

**Have all data underlying the figures and results presented in the manuscript been provided?**

Reviewer #1: Yes

Reviewer #2: Yes

PLOS authors have the option to publish the peer review history of their article (what does this mean?). If published, this will include your full peer review and any attached files.

Reviewer #1: No

Reviewer #2: No

**Data Deposition**

http://datadryad.org/submit?journalID=pgenetics&manu=PGENETICS-D-22-01364R1

**Press Queries**

---

## [Editor Report · Acceptance letter]

31 Jul 2023

PGENETICS-D-22-01364R1 

Rege-1 promotes C. elegans survival by modulating IIS and TOR pathways 

Dear Dr TSAI, 

We are pleased to inform you that your manuscript entitled "Rege-1 promotes C. elegans survival by modulating IIS and TOR pathways" has been formally accepted for publication in PLOS Genetics! Your manuscript is now with our production department and you will be notified of the publication date in due course.

With kind regards,

Jazmin Toth

PLOS Genetics

On behalf of:
